# Cdc13 is predominant over Stn1 and Ten1 in preventing chromosome end fusions

Zhi-Jing Wu[1], Jia-Cheng Liu[1], Xin Man[1], Xin Gu[1], Ting-Yi Li[1], Chen Cai[1,2], Ming-Hong He[1], Yangyang Shao[3], Ning Lu[3], Xiaoli Xue[3], Zhongjun Qin[3], Jin-Qiu Zhou[1,2]*

[1]The State Key Laboratory of Molecular Biology, CAS Center for Excellence in Molecular Cell Science, Shanghai Institute of Biochemistry and Cell Biology, Chinese Academy of Sciences; University of Chinese Academy of Sciences, Shanghai, China; [2]School of Life Science and Technology, ShanghaiTech University, Shanghai, China; [3]Key Laboratory of Synthetic Biology, CAS Center for Excellence in Molecular Plant Sciences, Shanghai Institute of Plant Physiology and Ecology, Chinese Academy of Sciences; University of Chinese Academy of Sciences, Shanghai, China

**Abstract** Telomeres define the natural ends of eukaryotic chromosomes and are crucial for chromosomal stability. The budding yeast Cdc13, Stn1 and Ten1 proteins form a heterotrimeric complex, and the inactivation of any of its subunits leads to a uniformly lethal phenotype due to telomere deprotection. Although Cdc13, Stn1 and Ten1 seem to belong to an epistasis group, it remains unclear whether they function differently in telomere protection. Here, we employed the single-linear-chromosome yeast SY14, and surprisingly found that the deletion of *CDC13* leads to telomere erosion and intrachromosome end-to-end fusion, which depends on Rad52 but not Yku. Interestingly, the emergence frequency of survivors in the SY14 *cdc13Δ* mutant was ~29 fold higher than that in either the *stn1Δ* or *ten1Δ* mutant, demonstrating a predominant role of Cdc13 in inhibiting telomere fusion. Chromosomal fusion readily occurred in the telomerase-null SY14 strain, further verifying the default role of intact telomeres in inhibiting chromosome fusion.

**\*For correspondence:**
jqzhou@sibcb.ac.cn

**Competing interests:** The authors declare that no competing interests exist.

## Introduction

Telomeres, the native ends of eukaryotic linear chromosomes, are critical for the maintenance of genome stability. They protect chromosome ends from nuclease degradation, homologous recombination, and end-to-end fusions (*Bianchi and Shore, 2008*; *Malyavko et al., 2014*; *McEachern et al., 2000*; *Smogorzewska and de Lange, 2004*; *Wellinger and Zakian, 2012*). In *Saccharomyces cerevisiae*, telomere length is approximately $300 \pm 75$ bp, comprising double-stranded $TG_{1-3}/C_{1-3}A$ repeats and a 12–14 nt-long G-rich single-stranded tail (*Larrivée et al., 2004*; *Wellinger et al., 1993*; *Zakian, 1996*). The subtelomeric region consists of X and Y′ elements (*Chan and Tye, 1983a*). Y′ elements, located immediately internal to telomeric DNA, are present in zero to four tandem copies and fall into two classes, Y′ long (6.7 kb) and Y′ short (5.2 kb). Y′ elements exhibit homogeneity, showing at least 98% identity, even between different strains (*Chan and Tye, 1983a*; *Chan and Tye, 1983b*; *Louis and Haber, 1992*). In contrast, X elements, which are located centromere-proximal to telomeres or Y′ elements, are less conserved, with 8–18% divergence occurring between various X elements, and they are found in virtually all telomeres (*Louis and Haber, 1991*).

Because of the end-replication problem observed in the vast majority of eukaryotic cells (*Olovnikov, 1971*; *Watson, 1972*), telomeres progressively shorten at a rate of 3–5 nt per cell division in yeast, leading to chromosome end erosions in the absence of telomerase (*Lendvay et al., 1996*; *Lingner, 1997b*; *Marcand et al., 1999*; *Singer and Gottschling, 1994*). Telomerase is a

ribonucleoprotein complex consisting of the catalytic subunit Est2, the RNA moiety Tlc1, accessory factors Est1 and Est3 and the Pop1/Pop6/Pop7 proteins (*Lemieux et al., 2016*; *Lendvay et al., 1996*; *Lundblad and Szostak, 1989*; *Singer and Gottschling, 1994*). Although the deletion of either *EST1* or *EST3* leads to a defect in telomerase function in vivo, the core telomerase complex formed by the reverse transcriptase Est2 and the RNA template Tlc1 is sufficient to catalyze telomere addition in vitro (*Liao et al., 2005*; *Lingner et al., 1997a*). Est1 recruits telomerase to telomeres through its interaction with the ssDNA-binding protein Cdc13. Additionally, it promotes G-quadruplex formation to stimulate telomerase activity and provide telomere protection (*Tong et al., 2011*; *Zhang et al., 2010*). The function of Est3 is less clear, and it has been suggested to activate telomerase (*Talley et al., 2011*).

Although most of the cells lacking telomerase undergo senescence after 50–100 rounds of cell division, a few cells can maintain their telomeres via recombination-dependent mechanisms, yielding two types of survivors, referred to as Type I and Type II. Type I survivors exhibit massive amplification of subtelomeric Y' sequences with very short $TG_{1-3}$ repeats at the ends, while Type II survivors amplify $TG_{1-3}$ repeats and display a heterogeneous pattern of long terminal $TG_{1-3}$ tracts (*Lundblad and Blackburn, 1993*; *Teng and Zakian, 1999*). The formation of Type I and Type II survivors relies on the *RAD51* and *RAD50* gene epistasis groups, respectively. Thus, the deletion of both *RAD50* and *RAD51* almost eliminates survivor generation, as observed in *rad52Δ* mutants (*Chen et al., 2001*; *Le et al., 1999*).

Telomeres share many similarities with double-strand breaks (DSBs) and have the potential to trigger DNA damage responses. To counteract this, telomere capping proteins have evolved to distinguish normal chromosome ends from DNA break ends generated from accidental DNA damage. In budding yeast, the double-stranded $TG_{1-3}$ repeats and the G tail are bound by an array of Rap1 and Cdc13-Stn1-Ten1 (CST) complexes, respectively (*Bourns et al., 1998*; *Gilson et al., 1993*; *Hughes et al., 2000*; *Lin and Zakian, 1996*; *Nugent et al., 1996*; *Price et al., 2010*; *Ray and Runge, 1999*; *Tsukamoto et al., 2001*). Rap1 is required for cell viability and performs many distinct essential cellular tasks, such as transcription regulation and telomere protection (*Buck and Lieb, 2006*; *Diffley, 1992*; *Lieb et al., 2001*; *Pardo and Marcand, 2005*; *Shore, 1994*; *Shore and Nasmyth, 1987*; *Tomar et al., 2008*). Thus, the causes of lethality due to Rap1 deletion remain a mystery.

The CST complex specifically binds telomeric single-stranded DNA (ssDNA) and acts as a telomere-capping protein and telomerase regulator. This trimeric complex structurally resembles heterotrimeric replication protein A (RPA), a eukaryotic major ssDNA-binding protein, indicating that the CST complex may function as a specialized telomere-dedicated RPA-like complex (*Gao et al., 2007*). Cdc13, the major single-stranded telomere-binding protein, plays dual roles in telomere protection and telomerase recruitment (*Bourns et al., 1998*; *Hughes et al., 2000*; *Lin and Zakian, 1996*; *Nugent et al., 1996*; *Tsukamoto et al., 2001*). Cdc13 interacts with Est1 and regulates telomerase access to telomeres in late S phase (*Chen et al., 2018*; *Evans and Lundblad, 1999*; *Osterhage et al., 2006*; *Pennock et al., 2001*; *Qi and Zakian, 2000*; *Taggart et al., 2002*; *Wu and Zakian, 2011*), after which G-rich strand extension is limited by Stn1, whose binding site partially overlaps with that of Est1 (*Chandra et al., 2001*; *Wang et al., 2000*). The Stn1 and Ten1 subunits are recruited to telomere ends via direct interaction with Cdc13. Both Stn1 and Ten1 are reported to display a relatively weak telomeric DNA binding affinity and to exhibit Cdc13-independent functions (*Gao et al., 2007*; *Qian et al., 2010*; *Qian et al., 2009*). The amino terminus of Stn1 is sufficient for Ten1 binding, while its carboxyl terminus interacts with both Cdc13 and Pol12 (a subunit of the DNA Polα complex) (*Grossi et al., 2004*; *Petreaca et al., 2006*; *Puglisi et al., 2008*). In addition to limiting G-rich strand extension, Stn1 promotes the recruitment of Polα for lagging strand DNA replication (*Grossi et al., 2004*). Notably, many lines of evidence support the notion that Stn1 is the primary effector of telomere capping. For instance, either fusing the DNA-binding domain of Cdc13 to Stn1 or cooverexpressing Ten1 with a truncated form of Stn1 is sufficient to bypass the essential function of Cdc13 (*Pennock et al., 2001*; *Petreaca et al., 2006*). Although Ten1 remains less well characterized, it appears to promote the activity of Cdc13 (*Qian et al., 2009*). Nevertheless, Cdc13, Stn1 and Ten1 are all required for cell viability and telomere length regulation. Loss-of-function mutations in each subunit result in the accumulation of telomeric ssDNA and abnormal elongation of telomeres (*Garvik et al., 1995*; *Grandin et al., 2001b*; *Grandin et al., 1997*), suggesting that Cdc13, Stn1 and Ten1 are epistatic in telomere protection. Mammalian CST (CTC1-STN1-TEN1)

appears to have extratelomeric functions in DNA replication (*Price et al., 2010*; *Stewart et al., 2012*; *Wang et al., 2014*; *Wang et al., 2019*). It remains unclear whether *Saccharomyces cerevisiae* CST has functions in DNA replication other than its confined roles at telomeres. One clue concerning this question comes from the observation that the overexpression of Stn1 results in its mislocalization to a nontelomeric region where it interferes with replication fork integrity (*Gasparyan et al., 2009*). Due to the lethality of *CST* deletion, the investigation of its roles both in telomere regions and at other genomic loci is difficult.

We recently successively fused the 16 chromosomes of budding yeast via a CRISPR-Cas9 method (*Shao et al., 2019b*) and created single-chromosome yeast strains, designated SY14 and SY15, which contain a single linear or circular chromosome, respectively (*Shao et al., 2019a*; *Shao et al., 2018*). We have utilized these strains to revisit telomere-associated processes and made several intriguing findings that could never have been revealed in multiple-chromosome yeast.

## Results

### Successive passages of single-chromosome yeast strains SY14 and SY15

The single-linear-chromosome yeast strain SY14 and the single-circular-chromosome yeast strain SY15 were generated through CRISPR-Cas9-mediated successive chromosome fusions (*Shao et al., 2019a*; *Shao et al., 2018*; *Shao et al., 2019b*; *Figure 1A*). To examine whether the SY14 and SY15 strains can perpetually maintain their self-renewal capability, we streaked several clones of the SY14 and SY15 strains on YPD plates 63 times at intervals of two days (*Figure 1B and C*). The progeny colonies obtained after various re-streaks displayed the same size as the parental colonies. The growth rates of the subclones in liquid YPD medium were nearly identical to those of the original clones (*Figure 1D and E*). In addition, a pulse-field-gel electrophoresis (PFG) analysis revealed that the single chromosome in SY14 remained intact during the passages (*Figure 1—figure supplement 1*). Moreover, telomere Southern blotting assay showed that in all successively passaged clones, the telomere restriction fragment (TRF) was maintained at a relatively stable length ($\sim$1.3 ± 0.1 kb), comparable to that in the parental clone (*Figure 1F*). Fluctuations of telomere length were observed in different cell passages as well as different clones in the same passage, and the reason for the fluctuation was not clear. Nevertheless, these results indicate that the genomes of the SY14 and SY15 strains remain stable, providing useful tools for the investigation of telomere-associated processes.

### The essentiality of CST is attributed to its roles in telomere protection

The natural ends of linear chromosomes must avoid being detected as DNA breaks. This is achieved by the evolution of telomeric binding proteins that mask the chromosomal DNA ends from constitutive exposure. Rap1 and the CST complex are essential for telomere protection as well as cell viability. Given that Rap1 also acts as a general transcription regulator (*Buck and Lieb, 2006*; *Diffley, 1992*; *Lieb et al., 2001*; *Shore, 1994*; *Shore and Nasmyth, 1987*) and that the CST complex may exhibit extratelomeric functions (*Gasparyan et al., 2009*), we constructed the SY14/SY14$^a$ *rap1Δ RAP1*, SY14/SY14$^a$ *cdc13Δ CDC13*, SY14/SY14$^a$ *stn1Δ STN1* and SY14/SY14$^a$ *ten1Δ TEN1* diploid strains, in which one chromosomal copy of the *RAP1*, *CDC13*, *STN1*, or *TEN1* gene was deleted in association with the introduction of the pRS316-*RAP1*, pRS316-*CDC13*, pRS316-*STN1* or pRS316-*TEN1* plasmid, respectively (see Materials and methods). We then performed tetrad dissections to obtain the SY14 *rap1Δ* pRS316-*RAP1*, SY14 *cdc13Δ* pRS316-*CDC13*, SY14 *stn1Δ* pRS316-*STN1* and SY14 *ten1Δ* pRS316-*TEN1* haploid strains. We also constructed the SY15 *rap1Δ* pRS316-*RAP1*, SY15 *cdc13Δ* pRS316-*CDC13*, SY15 *stn1Δ* pRS316-*STN1* and SY15 *ten1Δ* pRS316-*TEN1* haploid strains by introducing a plasmid-borne wild-type gene and then deleting the chromosomal copy of the gene. Thereafter, the plasmid-borne wild-type genes were counterselected on 5′-fluoroorotic acid (5′-FOA) plates. The results showed that Rap1 was indispensable for the viability of the circular-chromosome yeast SY15 (*Figure 2A*), suggesting that Rap1 is essential not only for telomere protection but also for gene transcription, consistent with a previous report that the telomere binding of Rap1 is not required for its essential functions (*Alexander and Zakian, 2003*). In contrast, the lack of *CDC13*, *STN1* or *TEN1* led to the death of SY14 cells but did not affect the viability (*Figure 2B*) or the growth rate (*Figure 2C*) of the SY15 strain. These data indicate that the lethality of CST depletion is mainly caused by defects in telomere protection. However, there has been a debate about whether

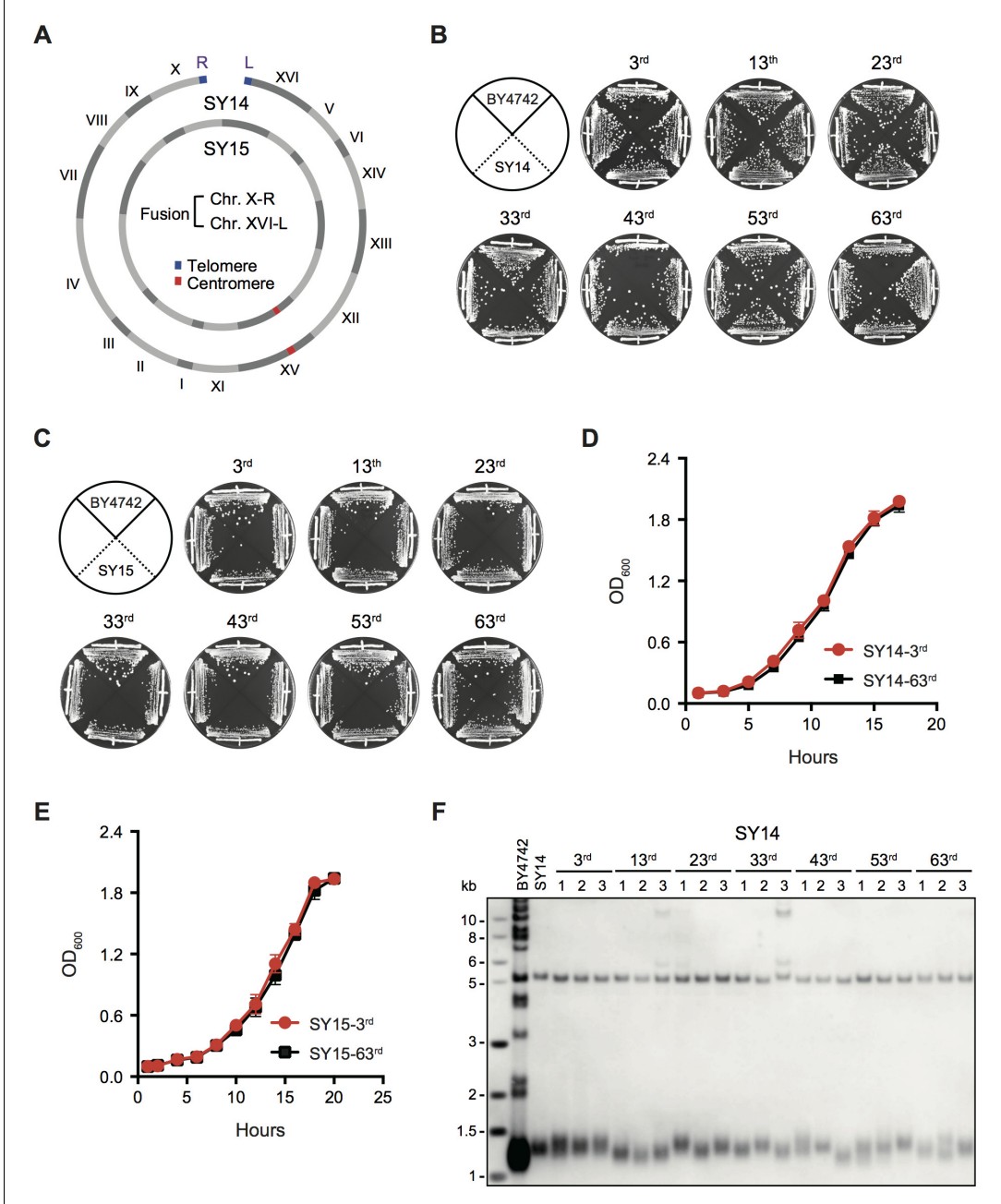

**Figure 1.** Successive passages of single-chromosome yeast strains SY14 and SY15 do not display growth change. (A) Schematic of single chromosome structure in yeast strains SY14 and SY15. Single linear and circular chromosomes of the SY14 and SY15 strains are respectively aligned in the outer and inner rings. The single circular chromosome of SY15 lacks telomeres of Chr X-R and Chr XVI-L in SY14. (B,C) Growth analysis of the SY14 (B) and SY15 (C) strains. Several clones of the SY14 and SY15 strains were re-streaked on YPD plates 63 times at intervals of two days. (D,E) Growth curves of the SY14 (D) and SY15 (E) clones at the 3rd and 63rd re-streaks. Error bars represent standard deviation (s.d.), n = 3. (F) Telomere southern blotting assay of the SY14 cells at different passages (labeled on top). At each passage, three independent clones were examined. The genomic DNA of the SY14 cells was digested by XhoI and subjected to Southern hybridization with a telomere-specific $TG_{1-3}$ probe.

The online version of this article includes the following source data and figure supplement(s) for figure 1:

**Source data 1.** Growth analysis of the SY14 and SY15 clones at the 3rd and 63rd re-streaks.

**Figure supplement 1.** The single chromosome in SY14 remains intact during the passages.

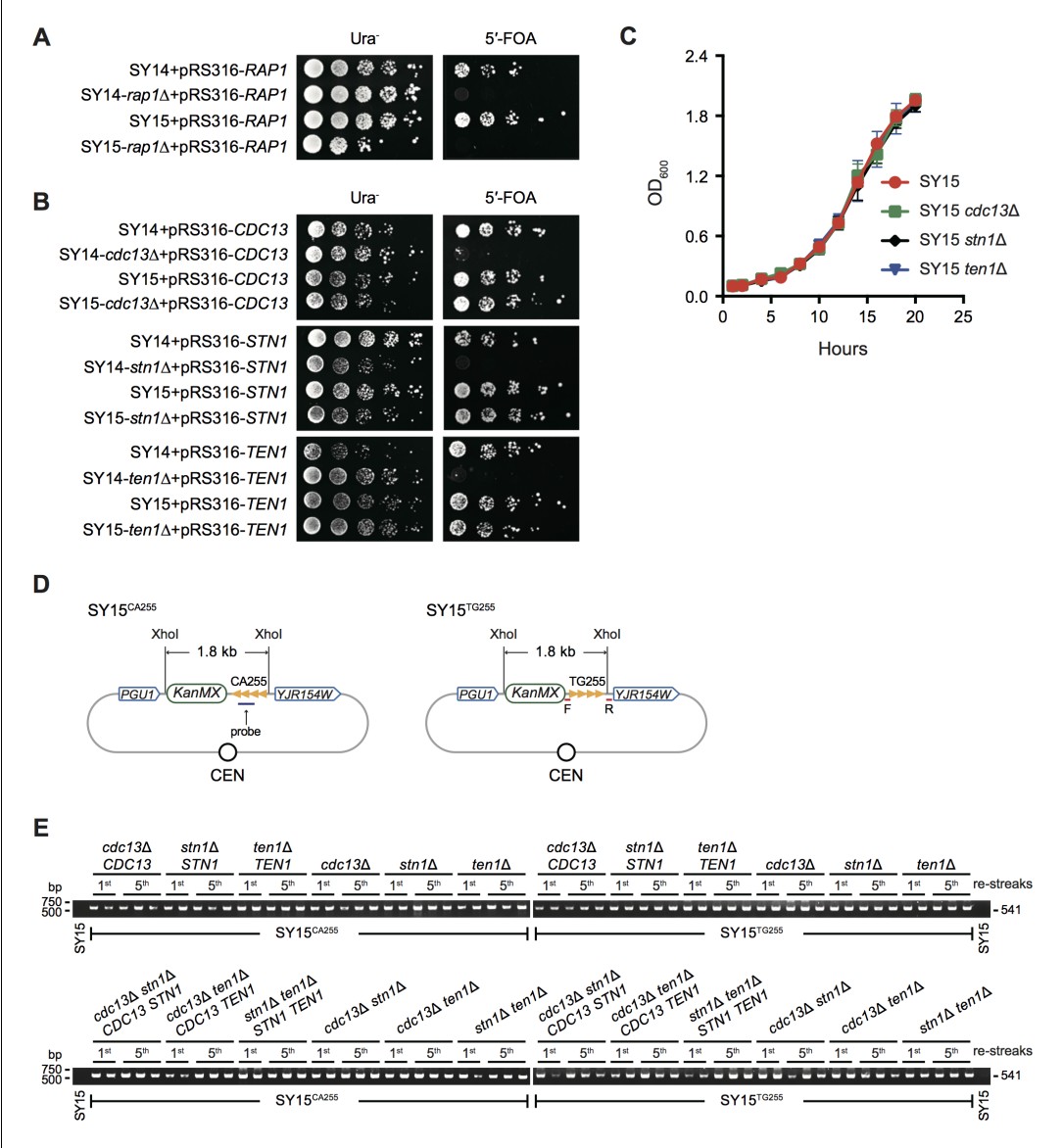

**Figure 2.** CST complex is not essential in single-circular-chromosome yeast strain SY15. (A) Growth analysis of *rap1Δ* mutants. SY14 *rap1Δ RAP1* and SY15 *rap1Δ RAP1* cells were spotted with five-fold dilutions on Ura⁻ medium (left panel) to select for the presence of pRS316-*RAP1* plasmid, and on 5′-FOA medium (right panel) to select for eviction of pRS316-*RAP1* plasmid. (B) Cell growth assay of *cdc13Δ*, *stn1Δ* and *ten1Δ* cells. SY14 *cdc13Δ CDC13*, SY15 *cdc13Δ CDC13*, SY14 *stn1Δ STN1*, SY15 *stn1Δ STN1*, SY14 *ten1Δ TEN1* and SY15 *ten1Δ TEN1* cells were spotted with five-fold dilutions on Ura⁻ medium (left panel) to select for the presence of pRS316 plasmids that contained a wild-type CST gene, and on 5′-FOA medium (right panel) to select for eviction of pRS316 plasmids. (C) Growth analysis of SY15 *cdc13Δ*, SY15 *stn1Δ* and SY15 *ten1Δ* cells. Three clones of each strain were examined. Error bars represent standard deviation (s.d.), n = 3. (D) Schematic representation of TG255/CA255 sequence insertion in the genomic loci of SY15^CA255 and SY15^TG255 strains. In single-circular-chromosome yeast SY15, a 255 bp telomeric sequence in both orientations (named CA255 or TG255, tandem orange triangles) is inserted between the *PGU1* and *YJR154W* genes. The TG probe for Southern blotting (*Figure 2—figure supplement 1*) and primer pairs for PCR-sequencing (E) are indicated in purple and red, respectively. The *KanMX* gene serves as a genetic marker for the integration of the telomeric tracts. The XhoI sites are used for restriction digestion in Southern blotting examining the insertions, which were ~1.8 kb in both SY15^CA255 and SY15^TG255 strains. This figure is not precisely drawn to scale. (E) Analysis of non-terminal telomere sequences by PCR. SY15^CA255 and SY15^TG255 strains (indicated at the bottom of each panel) were passaged on plates five times at intervals of two days. The genomic DNA was isolated from the 1st and 5th re-streaks (labeled at the top of each panel). Primers (5′-TCGACATCATCTGCCCAGAT-3′ and 5′-AGTTCGAACTAGGGTAATTG-3′) were used to amplify the DNA fragments flanking inserted telomeric sequence, and the PCR products were examined on agarose gels. Two or three independent clones of each genotype were examined.

The online version of this article includes the following source data and figure supplement(s) for figure 2:

**Source data 1.** Growth analysis of the SY15 *cdc13Δ*, SY15 *stn1Δ* and SY15 *ten1Δ* cells.

*Figure 2 continued on next page*

*Figure 2 continued*

**Figure supplement 1.** Southern blotting to determine the insertion of TG255/CA255 sequence in SY15^CA255 and SY15^TG255 strains.
**Figure supplement 2.** The full sequences of the clones of SY15^CA255 *cdc13Δ* (**A**) and SY15^TG255 *cdc13Δ* (**B**) cells at different passages were shown as representatives.

the CST complex functions like RPA in the replication of telomere DNA sequences, in addition to telomere protection (*Gao et al., 2007*; *Gasparyan et al., 2009*; *Price et al., 2010*; *Stewart, 2018*). The single-circular-chromosome yeast strains SY15 *cdc13Δ CDC13*, SY15 *stn1Δ STN1*, SY15 *ten1Δ TEN1*, SY15 *cdc13Δ stn1Δ CDC13 STN1*, SY15 *cdc13Δ ten1Δ CDC13 TEN1* and SY15 *stn1Δ ten1Δ STN1 TEN1* (SY15 *cstΔ CST*) appeared to be ideal tools for addressing this issue. Hence, we inserted a 255 bp-long telomeric sequence into a genomic locus between the *PGU1* and *YJR154W* genes in both orientations in the SY15 *cstΔ CST* strains. The resultant strains, which were designated SY15^CA255 and SY15^TG255, respectively (*Figure 2D*), were passaged on plates five times (~100 population doublings). The interstitial telomeric sequences (ITSs) of TG255/CA255 determined by Southern blotting were successfully transmitted from generation to generation in all the SY15^CA255 and SY15^TG255 strains, regardless of the presence or absence of *CST* (*Figure 2—figure supplement 1*). To determine whether there was any miniscule contraction or expansion of ITSs after ~100 rounds of replication, we performed PCR analysis, and the amplified DNA fragments from all of the strains were the same size (*Figure 2E*). Further sequencing results for three independent clones of each *cst* mutant confirmed that there were no mutations after ~100 population doublings (*Figure 2—figure supplement 2*). These results suggest that the CST complex does not affect the replication of ITSs; that is, the CST complex might have no function in telomere replication other than the recruitment of Polα for lagging strand synthesis (*Grossi et al., 2004*). However, it should be noted that the replication of ITSs may encounter difficulties in the absence of CST (*Gasparyan et al., 2009*; *Price et al., 2010*; *Stewart et al., 2012*; *Wang et al., 2014*; *Wang et al., 2019*), which leads to cell death and might not be recovered in this system. Additionally, or alternatively, the frequency of the expansion/contraction of ITSs in *cst* mutants was too low to be detected within ~100 rounds of replication (*Aksenova et al., 2015*).

## SY14 cells lacking CST survive by chromosome circularization

Although the deletion of a single CST component resulted in death of the majority of SY14 cells, sporadic clones could be seen on 5′-FOA plates (*Figure 2B*), indicating that some of the CST null cells escaped the fate of death and survived, which was not seen in the multiple-chromosome cells (i.e., BY4742 CST null strains). To verify the telomere structure of the survivors derived from SY14 *cdc13Δ* cells, we performed telomere Southern blotting to probe the telomeric TG$_{1-3}$ sequence. Notably, the Y′-telomere band of ~1.3 kb was not detected, while an ~15 kb band emerged in the majority of cells (27 out of 30) (*Figure 3A*), suggesting that the telomere structure of SY14 *cdc13Δ* survivors is totally different from that of SY14 cells. In SY14 cells, there are only two telomeres, which originated from Chr XVI-L and Chr X-R. The left-arm telomere of Chr XVI-L contains one copy of the Y′ and X elements, while the right-arm telomere of Chr X-R contains X but no Y′-element (*Figure 3B*). Presumably, the two chromosome ends could fuse together in SY14 *cdc13Δ* cells, as observed in fission yeast cells that lack Pot1, Stn1, Ten1 or Trt1 (*Baumann and Cech, 2001*; *Martín et al., 2007*; *Nakamura et al., 1998*; *Wang and Baumann, 2008*). Additionally, the hybridization signals detected in SY14 *cdc13Δ* survivors were quite similar to those of the SY15 cells (*Figure 3A*), pointing to the possibility that the eroded chromosome ends may have fused together in SY14 *cdc13Δ* cells.

To test this hypothesis and determine how much DNA had been lost prior to a potential fusion, we first performed PCR mapping to define the borders of telomere erosion in 74 independent clones (including the clones shown in *Figure 3A*) of the survivors. Primer pairs 1^L to 30^L covered the ~35 kb subtelomeric region of Chr XVI-L (*Figure 3—figure supplement 1A*), while primer pairs 1^R to 35^R covered the ~50 kb subtelomeric region of Chr X-R in intervals of ~1 kb (*Figure 3—figure supplement 1B*). The sequences of the primers are listed in *Supplementary file 1*. If a given pair of primers produced the correct PCR product, the corresponding chromosome region was considered to be intact; otherwise, it was considered to be lost. In the majority of cells (68 out of 74), the

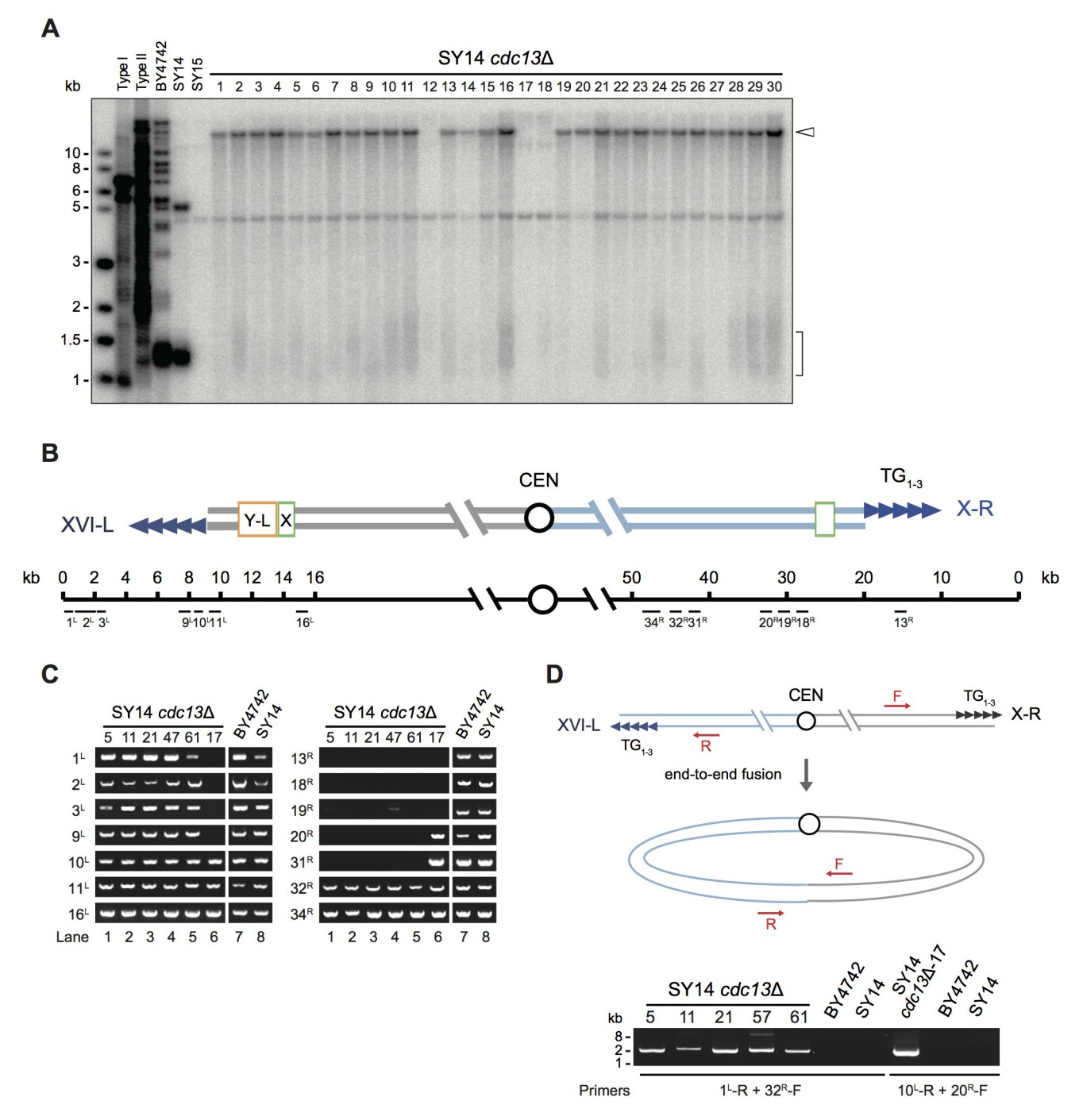

**Figure 3.** Survivors of SY14 *cdc13Δ* mutant contain a circularized chromosome. (**A**) Telomere Southern blotting analysis. 30 independent SY14 *cdc13Δ* colonies (labeled on top) were randomly picked, and their DNA was subjected to a telomere Southern blotting analysis to examine telomere structure. The bracket indicates Y′ telomere signals and the open arrowhead indicates the band of ~15 kb emerged in most of the clones except clones 12, 17 and 18. (**B**) Schematic representation of two chromosome arms of XVI-L and X-R in SY14 strain. Boxes in light green and yellow adjacent to telomeres (tandem blue triangles) represent subtelomeric X element and Y′-L element respectively. The numbers above the schematic line (chromosome) indicate the distance to the corresponding telomeric $TG_{1-3}$ sequences of XVI-L and X-R (not in precise scale). Black bars labeled $1^L$-$16^L$ or $13^R$-$34^R$ (under the schematic line) indicate the position of PCR primers that were used to examine either chromosomal end erosion. (**C**) Examples of PCR mapping results that define the borders of telomere erosion in SY14 *cdc13Δ* survivors. The primer pairs (shown in (**B**)) are indicated on left in each panel. The clone numbers of SY14 *cdc13Δ* are indicated on top in each panel. Primer sequences are listed in *Supplementary file 1*. (**D**) PCR examination of

*Figure 3 continued on next page*

*Figure 3 continued*

chromosome end-to-end fusion. Different pairs of primers (indicated at the bottom) were used to amplify the DNA fragments flanking the fusion points. The clone numbers of SY14 *cdc13Δ* are indicated on top.

The online version of this article includes the following figure supplement(s) for figure 3:

**Figure supplement 1.** Borders of erosion and rTG Type of SY14 *cdc13Δ* survivors are defined by mapping and PCR amplification.
**Figure supplement 2.** Fusion junctions in most SY14 *cdc13Δ* survivors contain TG sequences (rTG Type).
**Figure supplement 3.** Fusion junctions in most SY14 *cdc13Δ* survivors contain TG sequences (rTG Type).
**Figure supplement 4.** Fusion junction sequences of non-TG Type survivors derived from SY14 *cdc13Δ* mutants.

---

subtelomeric sequences of Chr XVI-L were retained (*Figure 3—figure supplement 1A*), while the sequences located approximately 43.5 kb away from the chromosome end of Chr X-R were lost (*Figure 3—figure supplement 1B*). Some of the typical PCR results are shown in *Figure 3C*. For example, in the *cdc13Δ*−5,–11, −21,–47 and −61 clones, PCR products were obtained using all primer pairs from $1^L$ to $16^L$ (for Chr XVI-L) (*Figure 3C*, left panel, lanes 1 to 5); PCR products were also obtained with primer pairs $32^R$ and $34^R$ but not with primer pairs $13^R$ to $31^R$ (for Chr X-R) (*Figure 3C*, right panel, lanes 1 to 5). These results indicated that in these clones, the chromosomal region of ~0.1 kb proximal to telomere XVI-L was intact, while the chromosomal region of ≤43.5 kb proximal to telomere X-R was lost. In wild-type BY4742 and SY14 cells, which were positive controls, all the primer pairs amplified PCR products of the predicted length (*Figure 3C*, lanes 7 and 8 in both panels).

The borders of telomere erosion in clones 12, 17 and 18 were likely different from those of most clones (*Figure 3—figure supplement 1*). For example, for the left chromosome arm (Chr XVI-L) in clone 17, PCR products were detected with primer pairs $10^L$ to $16^L$ but not with primers $1^L$ to $9^L$ (*Figure 3C*, left panel, lane 6), indicating that the chromosomal region of ~7.5 kb proximal to telomere XVI-L was lost. For the right chromosome arm (Chr X-R) in clone 17, PCR products were detected with primer pairs $20^R$ to $34^R$ but not with primers $13^R$ to $19^R$ (*Figure 3C*, right panel, lane 6), indicating that the chromosomal region of ~30.0 kb proximal to telomere X-R was lost.

To further examine whether chromosome fusion took place, we performed PCR analyses with pairs of primers targeting the ends of the two chromosome arms (*Figure 3D*, top panel). Presumably, a PCR product would be obtained only if the chromosome ends had fused together. The results showed that the primer pair consisting of $1^L$-R and $32^R$-F amplified DNA fragments with a size of ~2 kb in most of the survivor clones (*Figure 3—figure supplement 1C*), including *cdc13Δ*−5,–11, −21,–47 and −61, as shown in *Figure 3D* (bottom panel), while the primer pair consisting of $10^L$-R and $20^R$-F amplified a DNA fragment of ~1.8 kb in clone 17 (*Figure 3D*, bottom panel). These PCR products were not detected in the linear-chromosome yeast strains of either BY4742 or SY14. These results strongly support the hypothesis of chromosome fusion in SY14 *cdc13Δ* survivors.

To validate chromosome fusion and determine fusion points/sequences in SY14 *cdc13Δ* survivors, we cloned and sequenced all the PCR products from 71 clones. The sequencing results are summarized in *Figure 3—figure supplements 2–4*. Notably, in most survived clones, fusion took place in the telomeric 5′-TG$_{1-3}$-3′ region of Chr XVI-L and the distal '5′-(CA)$_{17}$-3'' region, which is located 43.9 kb away from the telomeric TG$_{1-3}$ sequence of Chr X-R (e.g., SY14 *cdc13Δ*−30, *Figure 4A*). The retained TG$_{1-3}$ sequences of Chr XVI-L in individual circular chromosomes were of different lengths: ranging from 21 to 374 bp long, with an average length of 126 bp (*Figure 3—figure supplements 2* and *3*). These survivors were designated 'rTG Type' clones (*Figure 4A*). One of the rTG Type clones was successively passaged on plates, and the cells grew as robustly as the SY15 cells (*Figure 4—figure supplement 1A*). Unlike the rTG Type clone, clones 12, 17 and 18 employed non-TG but homologous DNA sequences, which existed on both arms of the linear single chromosome of SY14, to mediate chromosome fusion (*Figure 4B* and *Figure 3—figure supplement 4*). These survivors were designated as non-TG Type clones (*Figure 4B*). Notably, in these non-TG Type clones, the fusion sequences differed, as did their lengths (26, 46 and 7 bp in clones 12, 17 and 18, respectively) (*Figure 4B* and *Figure 3—figure supplement 4*), suggesting sporadic events of chromosome fusions. The fusion sequences from the two arms of the linear chromosome were not 100% homologous (*Figure 4B* and *Figure 3—figure supplement 4*). For example, in clone 18, there was a T-C

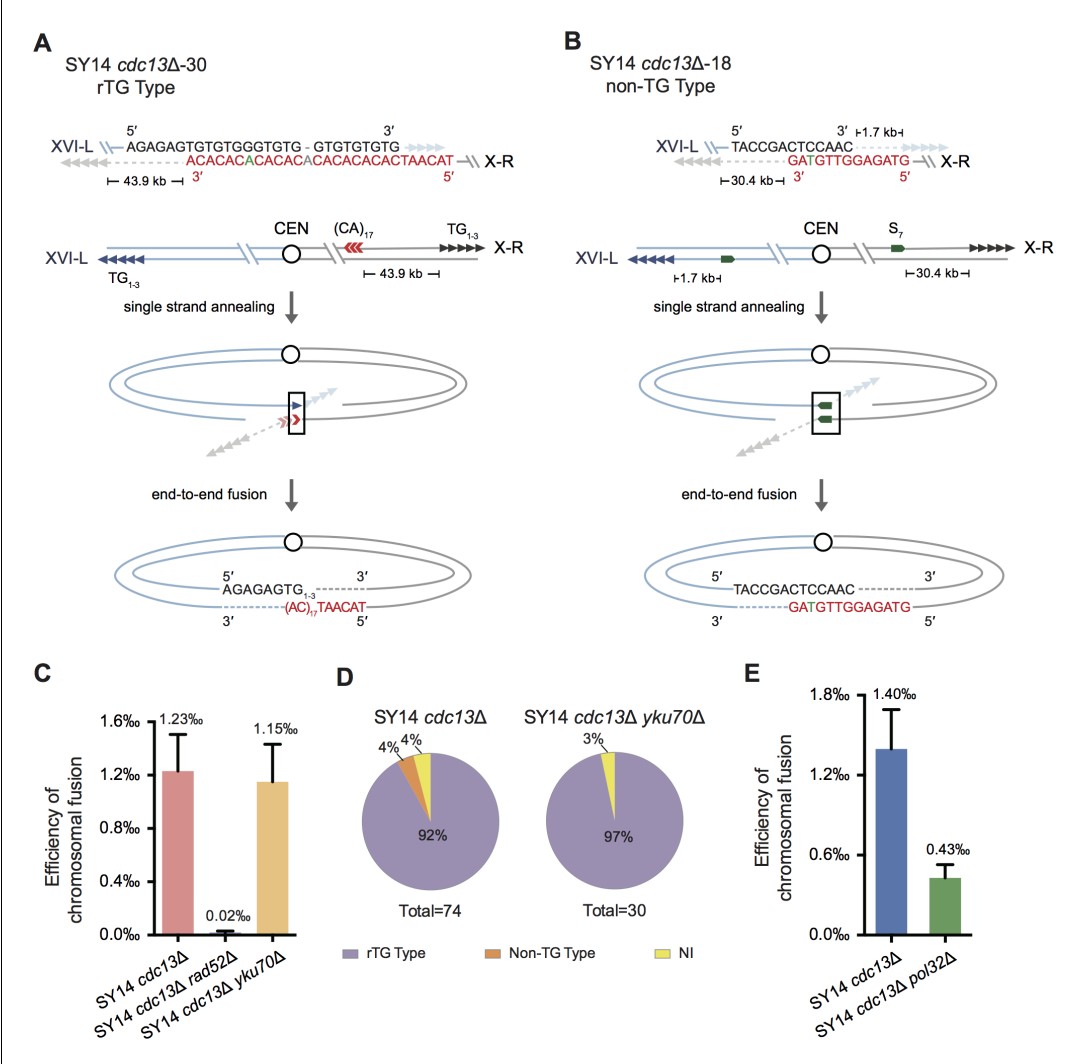

**Figure 4.** Chromosome fusion of SY14 *cdc13Δ* cells is nearly eliminated in the absence of Rad52. (**A**) Schematic of rTG Type survivors in SY14 *cdc13Δ*. In SY14 *cdc13Δ* clone 30, the fusion region of TG$_{1-3}$ sequence (in black) is in Chr XVI-L, and the (CA)$_{17}$ region (in red) locates 43.9 kb away from of Chr X-R. Bases in green are mis-paired, bases in grey or dashes are deleted. (**B**) Schematic of non-TG Type survivors in SY14 *cdc13Δ*. In SY14 *cdc13Δ* clone 18, the fusion sequence of CTCCAAC (in black) is 1.7 kb away from Chr XVI-L telomere, and the fusion sequence of GTTGTAG (in red) is 30.4 kb away from of Chr X-R telomere. Bases in green are mis-paired. (**C**) Quantification of survivor generation rates of SY14 *cdc13Δ* (1.23‰), SY14 *cdc13Δ rad52Δ* (0.02‰) and SY14 *cdc13Δ yku70Δ* (1.15‰) cells. Error bars represent standard deviation (s.d.), n = 3. (**D**) Percentage of rTG Type, non-TG Type and not-identified (NI) survivors in SY14 *cdc13Δ* (n = 74) and SY14 *cdc13Δ yku70Δ* (n = 30) strains. (**E**) Quantification of survivor generation rates of SY14 *cdc13Δ* (1.40‰) and SY14 *cdc13Δ pol32Δ* (0.43‰) cells. Error bars represent standard deviation (s.d.), n = 3.

The online version of this article includes the following source data and figure supplement(s) for figure 4:

**Source data 1.** Quantification of survivor generation rates of SY14 *cdc13Δ*, SY14 *cdc13Δ rad52Δ*, SY14 *cdc13Δ yku70Δ* and SY14 *cdc13Δ pol32Δ* cells.
**Source data 2.** NHEJ efficiency of SY14, SY14 *lig4Δ* and SY14 *yku70Δ* strains.
**Figure supplement 1.** Survivors harboring circular chromosome maintain a stable genome.
**Figure supplement 2.** Borders of erosion (A and B) and rTG Type (C) of SY14 *cdc13Δ yku70Δ* survivors are defined by mapping and PCR amplification.
**Figure supplement 3.** Fusion junctions of rTG Type in SY14 *cdc13Δ yku70Δ* survivors.
**Figure supplement 4.** NHEJ pathway is still functional in single-linear-chromosome yeast SY14.

mismatch at the fusion point (*Figure 4B*). Nevertheless, these sequencing results were consistent with the results of the telomere Southern blotting assay (*Figure 3A*): a 15 kb band emerged in all rTG Type cells, while no 15 kb band was detected in non-TG Type cells.

For clones 32, 45 and 68, the borders of erosion were identified (*Figure 3—figure supplement 1A and B*), but the fusion sequences could not be defined due to failure of PCR amplification. The

reason for this failure was not clear. One possibility was that the fusion segments were too large to be amplified due to unexpected insertions at the junctions. These clones were classified as NI (not identified).

## Cdc13 inhibits Rad52-mediated chromosomal end fusion

Since the chromosome end fusions involved homologous sequences (e.g., the $TG_{1-3}/C_{1-3}A$ sequence in rTG Type survivors) buried within both chromosome ends, we wanted to know whether the fusion of single chromosome ends depended on the *RAD52* pathway, which is essential for homologous recombination (*San Filippo et al., 2008*; *Symington, 2002*). To this end, we deleted *RAD52* or *YKU70* in the SY14 *cdc13Δ CDC13* strain and performed a quantitative survivor formation assay. An aliquot of the cell cultures was plated on either 5′-FOA or yeast complete (YC) medium lacking uracil (Ura⁻) to select for the eviction or the presence of plasmid-borne wild-type *CDC13* genes. Colonies were counted, and the efficiency of chromosomal fusion was measured by dividing the number of cells on 5′-FOA by that on YC medium lacking uracil (Ura⁻). The results revealed that the efficiency of chromosomal fusion in the SY14 *cdc13Δ rad52Δ* strain was over 60-fold lower than that in the SY14 *cdc13Δ* strain (*Figure 4C*), indicating that chromosome-end fusions were mainly mediated by the Rad52 pathway. In contrast, the efficiency of chromosomal fusion in the SY14 *cdc13Δ yku70Δ* strain was comparable to that in the SY14 *cdc13Δ* strain (*Figure 4C*), suggesting that the NHEJ pathway contributes little to chromosomal fusion. Additionally, we characterized chromosome fusion in SY14 *cdc13Δ yku70Δ* survivors. As in SY14 *cdc13Δ* survivors, we randomly picked 30 single colonies and determined the borders of erosion (*Figure 4—figure supplement 2A and B*), fusion types (*Figure 4—figure supplement 2C*) and fusion junction sequences (*Figure 4—figure supplement 3*). The colonies that could produce an ~2 kb band using the 1 $^L$-R and 32 $^R$-F primer pairs were categorized as rTG Type colonies, and further DNA sequencing results confirmed that 29 clones (out of 30) were rTG Type colonies (*Figure 4—figure supplements 2C* and *3*). Only one clone produced no PCR products with primers oriented from the centromere to the telomere, though the borders of erosion were determined. These results indicated that the rTG Type survivor generation rate in SY14 *cdc13Δ yku70Δ* cells was comparable to that in SY14 *cdc13Δ* cells (*Figure 4D*), further supporting the notion that two chromosome ends fused together in SY14 *cdc13Δ* cells, likely via the Rad52 pathway (homologous recombination). However, it remained possible that the predominant Rad52-mediated repair of deprotected telomeres in the SY14 *cdc13Δ* strain was a result of defective NHEJ activity. To test this hypothesis, we deleted *YKU70* as well as *LIG4* in the SY14 strain and employed a plasmid repair assay as previously described (*Boulton and Jackson, 1996*; *Wilson and Lieber, 1999*; *Zhang and Paull, 2005*) in the SY14 *lig4Δ*, SY14 *yku70Δ* and SY14 strains (*Figure 4—figure supplement 4A*). Compared with the SY14 *lig4Δ* and SY14 *yku70Δ* strains, in which the NHEJ pathway was blocked, the NHEJ pathway in the SY14 strain functioned efficiently (*Figure 4—figure supplement 4B*).

Pol32 is involved in both break-induced replication (BIR) and microhomology-mediated end joining (MMEJ) pathways (*Lee and Lee, 2007*; *Lydeard et al., 2007*) and is required for survivor generation in telomerase-null BY4742 cells. To further investigate whether chromosomal circularization in SY14 *cdc13Δ* cells relies on Pol32, we deleted *POL32* in SY14 *cdc13Δ* pRS316-*CDC13* cells. The quantitative survivor formation assay showed that the further deletion of *POL32* in the *cdc13Δ* mutant resulted in an approximately 3-fold decrease in chromosome fusion rates compared to the single deletion of *CDC13* (*Figure 4E*), indicating that chromosomal circularization in SY14 *cdc13Δ* survivors partially depends on Pol32.

## The deletion of either *STN1* or *TEN1* results in a dramatic reduction in the chromosome fusion frequency compared to the deletion of *CDC13*

*CDC13*, *STN1* and *TEN1* are essential genes, and their single deletion mutants are inviable (*Garvik et al., 1995*; *Grandin et al., 2001b*; *Grandin et al., 1997*). Loss-of-function alleles of each subunit result in the accumulation of telomeric ssDNA and the abnormal elongation of telomeres (*Garvik et al., 1995*; *Grandin et al., 2001b*; *Grandin et al., 1997*). These lines of evidence suggest that Cdc13, Stn1 and Ten1 belong to a single epistatic group involved in telomere protection. To test this hypothesis further, we assessed the frequency of chromosome fusions in SY14 *stn1Δ* and *ten1Δ* cells. The results showed that the frequency of intrachromosomal fusion in the SY14 *stn1Δ* and

SY14 *ten1Δ* strains was 0.04‰, which was ~29 fold lower than the 1.14‰ frequency observed in the *cdc13Δ* strain (*Figure 5A*), indicating that Cdc13, Stn1 and Ten1 play nonequivalent roles in telomere capping: Cdc13 plays a much more dominant role than Stn1 and Ten1 in inhibiting chromosome end fusions (*Figure 5A*). The further mapping of chromosome erosion (*Figure 5—figure supplements 1* and *2*), the determination of chromosome fusions and the characterization of the

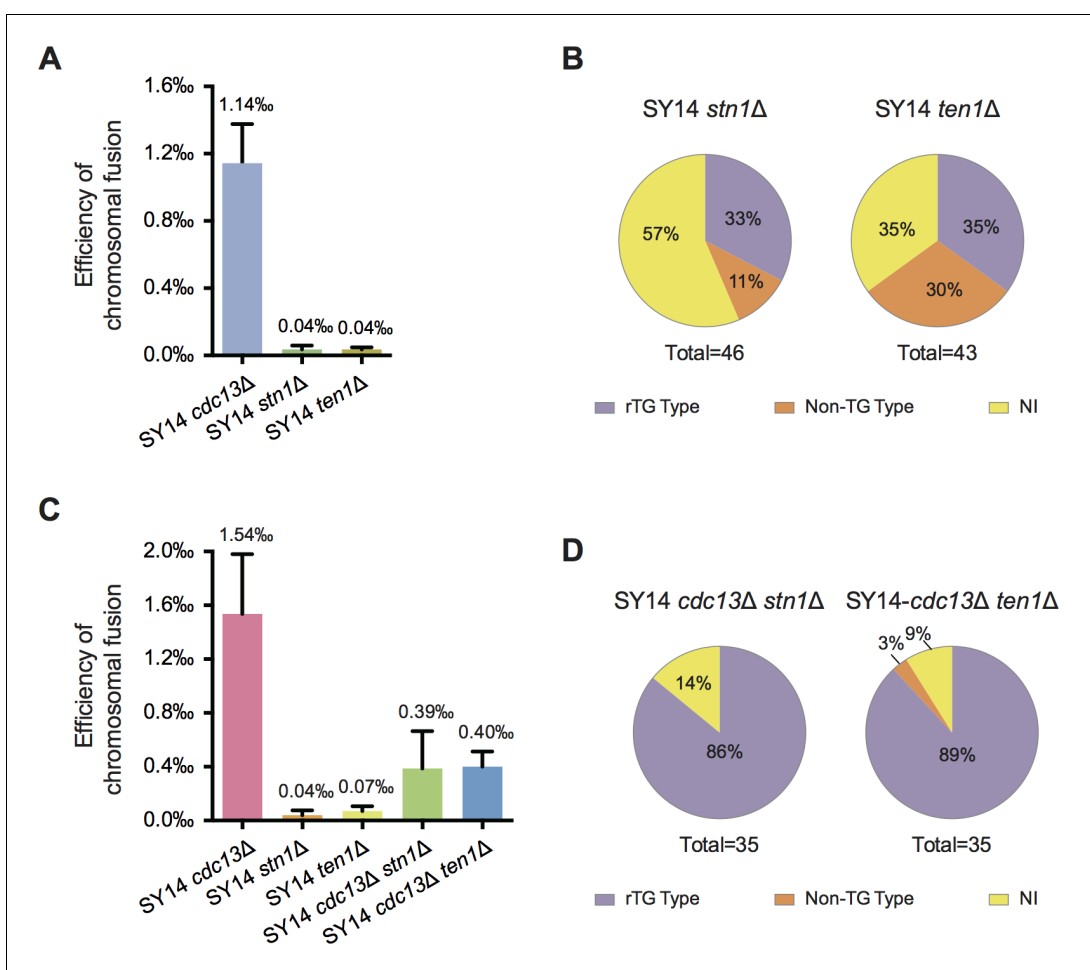

**Figure 5.** Chromosome fusion frequency in either SY14 *stn1Δ* or SY14 *ten1Δ* cells is much lower than that in SY14 *cdc13Δ* cells. (A) Quantification of survivor generation rates of SY14 *cdc13Δ* (1.14‰), SY14 *stn1Δ* (0.04‰) and SY14 *ten1Δ* (0.04‰) cells. Error bars represent standard deviation (s.d.), n = 3. (B) Percentage of rTG Type, non-TG Type and not-identified (NI) survivors in SY14 *stn1Δ* (n = 46) and SY14 *ten1Δ* (n = 43) cells. (C) Quantification of survivor generation rates of SY14 *cdc13Δ* (1.54‰), SY14 *stn1Δ* (0.04‰), SY14 *ten1Δ* (0.07‰), SY14 *cdc13Δ stn1Δ* (0.39‰) and SY14 *cdc13Δ ten1Δ* (0.40‰) cells. Error bars represent standard deviation (s.d.), n = 3. (D) Percentage of rTG Type, non-TG Type and not-identified (NI) survivors in SY14 *cdc13Δ stn1Δ* (n = 35) and SY14 *cdc13Δ ten1Δ* (n = 35) cells. The online version of this article includes the following source data and figure supplement(s) for figure 5:

**Source data 1.** Quantification of survivor generation rates of SY14 *cdc13Δ*, SY14 *stn1Δ*, SY14 *ten1Δ*, SY14 *cdc13Δ stn1Δ* and SY14 *cdc13Δ ten1Δ* cells.

**Figure supplement 1.** PCR mapping of the borders of Chr XVI-L erosion in SY14 *stn1Δ* (A) and *ten1Δ* (B) survivors.

**Figure supplement 2.** PCR mapping of the borders of Chr X-R erosion in SY14 *stn1Δ* (A) and *ten1Δ* (B) survivors.

**Figure supplement 3.** Determination of rTG Type survivors by PCR (A and B) and fusion junction sequences of non-TG Type survivors (C) in SY14 *stn1Δ* and SY14 *ten1Δ* mutants.

**Figure supplement 4.** PCR mapping of the borders of Chr XVI-L erosion (A) and Chr X-R erosion (B) in SY14 *cdc13Δ stn1Δ* and *cdc13Δ ten1Δ* survivors.

**Figure supplement 5.** Determination of rTG Type survivors by PCR (A) in *cdc13Δ stn1Δ* and *cdc13Δ ten1Δ* survivors and fusion junction sequences of non-TG Type survivors (B) derived from SY14 *cdc13Δ ten1Δ* mutants.

**Figure supplement 6.** SY14^CA255 *cstΔ* survived cells utilize CA255 sequence for chromosomal circularization.

fusion types (*Figure 5—figure supplement 3A and B*) revealed that the frequency of rTG Type survivors (telomeric DNA sequence-mediated fusion) in SY14 *stn1Δ* and SY14 *ten1Δ* cells was ~30% (*Figure 5B*), which was much lower than the 92% frequency observed in SY14 *cdc13Δ* cells (*Figure 4D*). Accordingly, non-TG Type survivors were increased in SY14 *stn1Δ* and SY14 *ten1Δ* strains (*Figure 5B* and *Figure 5—figure supplement 3C*), consistently supporting the notion that in the absence of either Stn1 or Ten1, Cdc13 still binds telomeric ssDNA and prevents telomeric $TG_{1-3}$ sequence(s) from recognition by CA-rich (e.g., $(CA)_{17}$) sequences, thus inhibiting chromosome fusion.

If *CDC13* is epistatic to *STN1* or *TEN1* in inhibiting chromosome fusion, the frequency of chromosome fusion in the *cdc13Δ stn1Δ* or *cdc13Δ ten1Δ* double mutant should be similar to that in the *cdc13Δ* single mutant. Thus, we quantified the chromosome fusion frequency in *cdc13Δ stn1Δ* and *cdc13Δ ten1Δ* double mutants and found that the further deletion of *CDC13* in *stn1Δ* and *ten1Δ* mutants resulted in approximately 10- and 6-fold increases in chromosome fusion rates compared to the single deletion of *STN1* or *TEN1*, respectively (*Figure 5C*). Further detailed mapping (*Figure 5—figure supplement 4*) and characterization (*Figure 5—figure supplement 5*) of chromosome fusions revealed that the increase in chromosome fusion in *cdc13Δ stn1Δ* and *cdc13Δ ten1Δ* double mutants was likely attributed to the elevation of the percentage of rTG Type chromosome fusion, which was 86% and 89% in the *cdc13Δ stn1Δ* and *cdc13Δ ten1Δ* mutants, respectively (*Figure 5D*), comparable to the 92% rate in SY14 *cdc13Δ* cells (*Figure 4D*). In contrast, 0% (in *cdc13Δ stn1Δ*) and 3% (in *cdc13Δ ten1Δ*) of the total survivors were non-TG Type clones (*Figure 5D* and *Figure 5—figure supplement 5C*). These results further support the conclusion that the presence of Cdc13 in telomeric ssDNA inhibits chromosome fusion. Interestingly, the chromosome fusion rates in the *cdc13Δ stn1Δ* and *cdc13Δ ten1Δ* strains were significantly lower than those in the *cdc13Δ* strain, suggesting that independent of Cdc13, Stn1 and Ten1 play roles in inhibiting chromosome fusion. This observation is in line with previous reports showing that Stn1 and Ten1 shield chromosome ends from extensive resection by exonucleases, likely because they both bind telomeres with a relatively low affinity (*Gao et al., 2007*; *Qian et al., 2010*; *Qian et al., 2009*), and Stn1 functions as the primary effector of telomere protection (*Pennock et al., 2001*; *Petreaca et al., 2006*; *Petreaca et al., 2007*; *Sun et al., 2009*). Taken together, these results indicate that it is Cdc13, and not Stn1 or Ten1, that plays the major role in inhibiting chromosomal end-to-end fusion.

Presumably, a lack of Cdc13 protection results in an equal probability of telomere erosion at both ends of the single chromosome in SY14 cells. However, in *cdc13Δ* rTG Type survivors, the resection of two telomeres before circularization was relatively unbalanced (*Figure 4A*). We propose that telomere erosion occurred at the two telomeres with an equal probability, but the pre-existing 5′-$(CA)_{17}$-3′ sequence, which is located 43.9 kb away from the telomere of Chr X-R in SY14, provided an optimal complementary sequence to pair with the 5′-$TG_{1-3}$-3′ telomeric sequence of Chr XVI-L, thus favoring the generation (selection) of rTG Type survivors. To test this hypothesis, we constructed the SY14$^{CA255}$ strain, in which a 255 bp $C_{1-3}A$ sequence was inserted between the *PGU1* and *YJR154W* genes on Chr X-R (*Figure 5—figure supplement 6A*). The inserted $C_{1-3}A$ sequence was located 20.9 kb away from the telomeric $TG_{1-3}$ sequence of Chr X-R, more proximal than the $(CA)_{17}$ sequence to the telomere of Chr X-R. Likewise, the single or double deletion of the *CST* gene resulted in the death of the majority of SY14$^{CA255}$ cells, but some survivors emerged. A single clone of each isogenic survivor was characterized; that is the borders of the erosion and fusion types of these survivors were determined by PCR mapping (*Figure 5—figure supplement 6B–D*). The results revealed that all of the clones were of the rTG Type and utilized CA255 instead of the '5′-$(CA)_{17}$-3′' repeat sequence at their fusion junctions (*Figure 5—figure supplement 6E*). Therefore, chromosome circularization preferentially occurred at sites where complementary sequences existed on two telomeres. However, we could not exclude the possibility that different chromatin structures at the two telomeres also contributed to the unbalanced erosion.

## Telomerase inactivation in SY14 cells results in cellular senescence and chromosome circularization

Our previous study showed that a lack of telomerase in SY14 led to senescence and the generation of survivors, whose telomere structures were distinguished from those of canonical Type I and Type II survivors (*Shao et al., 2019a*). This phenotype brought to mind the notion that the eroded chromosome ends of SY14 *tlc1Δ* cells might also fuse together. Hence, we constructed the SY14 *tlc1Δ*

pRS316-*TLC1* and SY15 *tlc1Δ* pRS316-*TLC1* strains, in which the chromosomal copy of the *TLC1* gene was deleted, while a plasmid-borne wild-type *TLC1* gene (pRS316-*TLC1*) was introduced. The SY14 *tlc1Δ* pRS316-*TLC1* cells and SY15 *tlc1Δ* pRS316-*TLC1* cells were grown on plates containing 5′-FOA, which counterselected the clones that had lost the pRS316-*TLC1* plasmid. As a result, the SY14 *tlc1Δ* strain and SY15 *tlc1Δ* strain were obtained. The phenotypes of senescence and telomere length in the SY14 *tlc1Δ* and SY15 *tlc1Δ* strains were examined. In liquid-grown culture, three individual colonies senesced and recovered in different passages (*Figure 6A*). Southern blotting assay revealed that the telomeres of SY14 *tlc1Δ* cells shortened with increasing passages and reached a critical length at different time points when the cells were in the senescent state (*Figure 6B*), which is an indication of a correlation between telomere erosion and cellular senescence. Normally, Type II survivors grow faster than Type I survivors and eventually come to dominate liquid cultures. Among three individual clones, two (clones 1 and 3) generated Type II survivors with the amplification of TG repeats, while one (clone 2) exhibited a pattern that was totally different from that of either Type I or Type II survivors but was similar to that of SY14 *cdc13Δ* cells (*Figures 3A* and *6B*), suggesting that chromosome fusion had taken place. In solid medium, the SY14 *tlc1Δ* cells senesced at the 3rd re-streak (~75 generations) on plates (*Figure 6C*), and survivors gradually emerged thereafter. 50 independent clones of survivors were randomly selected, and telomere structure was determined by the telomere Southern blotting assay. The results showed that the telomere band of ~1.3 kb was not seen in all survivors, while a 15 kb band emerged in the majority of clones (*Figure 6D*), similar to the chromosomal end-to-end fusions observed in SY14 *cdc13Δ* survivors (*Figure 3A*). Further erosion-border mapping (*Figure 6—figure supplement 1*), fusion type assay (*Figure 6—figure supplement 2A*) and fusion junction sequencing (*Figure 6—figure supplements 2B* and *3*) confirmed that most of the survived SY14 *tlc1Δ* cells had experienced chromosomal circularization via homologous $TG_{1-3}$/$C_{1-3}A$ DNA sequences, as seen in SY14 *cdc13Δ* survivors, which could be perpetually maintained under laboratory conditions (*Figure 4—figure supplement 1B*). As expected, a 15 kb band was detected in all TG Type cells except for two clones, which contained $TG_{1-3}$/$C_{1-3}A$ tracts at the junction points that were too short (18 and 20 bp long). Accordingly, no corresponding band was detected in the non-TG Type and NI cells (*Figure 6D*; *Figure 6—figure supplements 1* and *2B*). Notably, in the rTG Type circular chromosome of SY14 *tlc1Δ* cells, the retained $TG_{1-3}$/$C_{1-3}A$ sequences at junction points were 18 to 58 bp long (36 bp long on average) (*Figure 6—figure supplement 3*), which was much shorter than those in SY14 *cdc13Δ* survivors. We propose that in SY14 *tlc1Δ* cells, chromosome fusion takes place passively after telomeres have experienced extensive shortening (crisis), while in SY14 *cdc13Δ* cells, chromosome fusion occurs immediately upon telomere deprotection via the loss of Cdc13.

Chromosome fusion takes place in fission yeast, which usually contain three chromosomes, in the absence of telomerase or telomere-binding proteins (*Baumann and Cech, 2001*; *Martín et al., 2007*; *Nakamura et al., 1998*; *Tashiro et al., 2017*; *Wang and Baumann, 2008*). Since SY14 *tlc1Δ* cells survived via chromosome fusion, we wanted to know whether chromosome fusions readily occurred in budding yeast with the decrease in chromosome numbers. We deleted *TLC1* in a series of chromosome-fused yeast strains, including SY1, SY3, SY5, SY7, SY8, SY9, SY10, SY11 and SY12, which were derived from the BY4742 strain and contained different numbers of chromosomes (*Figure 6—figure supplement 4A*; *Shao et al., 2018*). Fifteen single colonies of each *tlc1Δ* strain were immediately passaged on plates 7 to 10 times successively after *TLC1* deletion. Because not every strain displayed typical senescence and growth crisis, we assumed that passaging for such a long time allows the formation of survivors. The telomere pattern of these survivors was examined by Southern blotting. The results showed that BY4742 *tlc1Δ* generated both Type I and Type II survivors (*Lundblad and Blackburn, 1993*; *Teng et al., 2000*; *Teng and Zakian, 1999*). With the reduction in chromosome numbers, the frequency of Type II survivor emergence gradually decreased, while Y′ recombination was readily detected, indicating an increase of Type I survivors (*Figure 6—figure supplement 4B*-E). These data suggest that the efficiency of telomere recombination is affected by chromosomal numbers in budding yeast. Notably, some distinct bands (indicated by open arrows on the right in the panels) with a size larger or smaller than the Y′-element occasionally emerged in SY7 *tlc1Δ* clones (eight chromosomes) as well as in SY *tlc1Δ* clones with fewer chromosomes, suggesting that in yeast strains with eight or fewer chromosomes, chromosome fusions might exist. However, most telomerase-null cells likely prefer to utilize the canonical telomere recombination pathway rather than chromosome fusion to survive. The reasons for this are not clear. One possibility is that

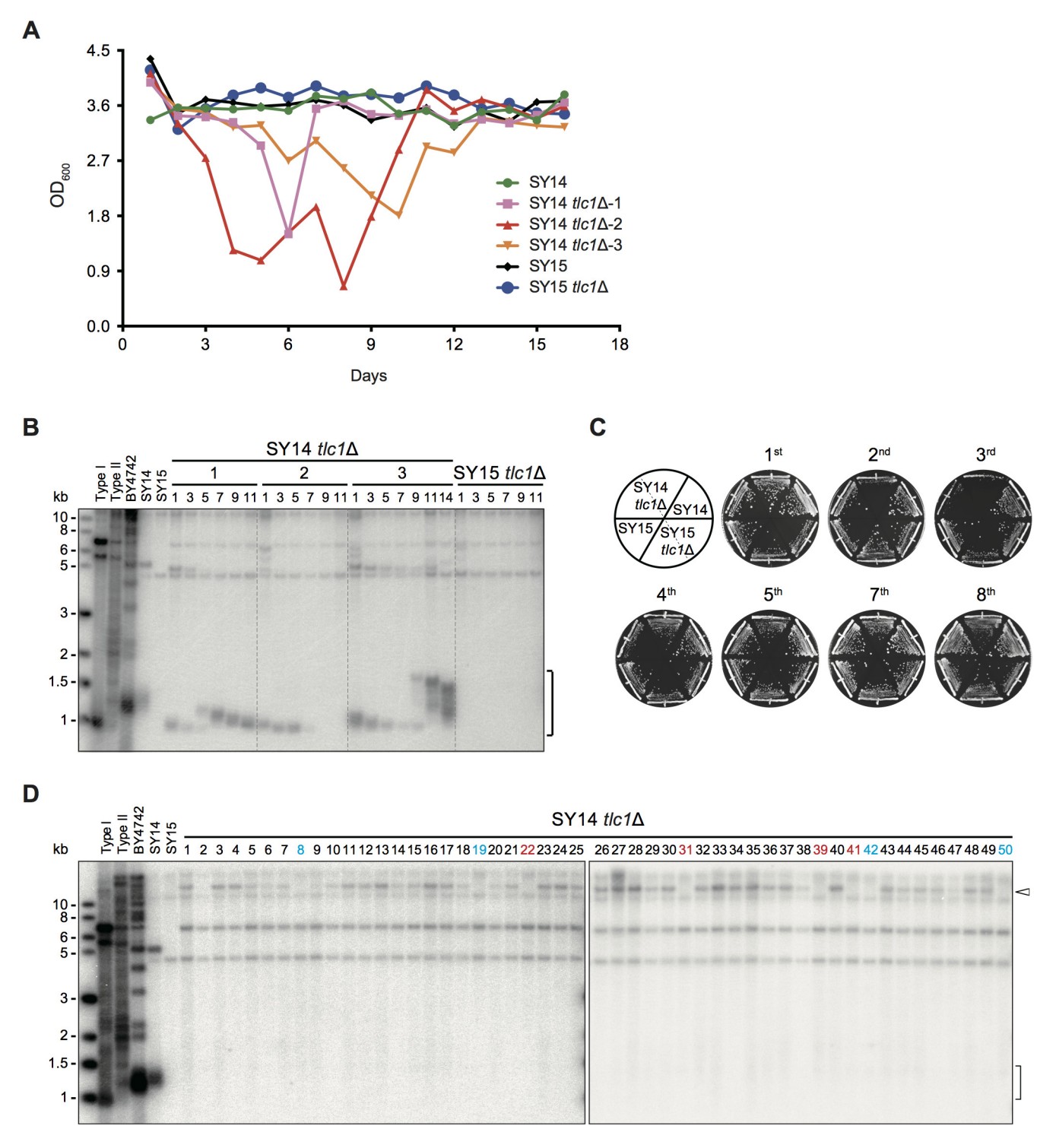

**Figure 6.** Telomerase inactivation in SY14 cells results in senescence and survivor formation. (**A**) Senescence assay in liquid medium. The growth (OD$_{600}$) of SY14 (green), SY14 *tlc1Δ* (three clones in pink, red and orange), SY15 (black) and SY15 *tlc1Δ* (blue) strains were monitored every 24 hr for 16 days. (**B**) Telomeric Southern blotting assay. Genomic DNA of the SY14 *tlc1Δ* and SY15 *tlc1Δ* strain examined in (**A**) were digested by XhoI and subjected to a Southern blotting analysis. The bracket indicates Y' telomere signals. (**C**) Senescence assay of the SY14 *tlc1Δ* and SY15 *tlc1Δ* strains on solid medium. After eviction of the pRS316-TLC1 plasmid in SY14 *tlc1Δ TLC1* or SY15 *tlc1Δ TLC1* strains by 5'-FOA selection, two independent SY14 *tlc1Δ* and SY15 *tlc1Δ* clones were re-streaked eight times to allow survivors to form. SY14 and SY15 were controls. (**D**) Telomere Southern blotting analysis of SY14 *tlc1Δ* survivors obtained on solid medium. 50 independent survivor clones (labeled 1 to 50 on top) were randomly picked, and their

*Figure 6 continued on next page*

*Figure 6 continued*

genomic DNA was subjected to Southern blotting assay with a telomeric $TG_{1-3}$ probe. The clone numbers in red are non-TG Type survivors. The clone numbers in blue are not-identified survivors. The bracket indicates Y' telomere signals and the open arrowhead indicates the new band of ~15 kb emerged in the majority of survivors.

The online version of this article includes the following source data and figure supplement(s) for figure 6:

**Source data 1.** Senescence assay of SY14 *tlc1*Δ and SY15 *tlc1*Δ cells.
**Figure supplement 1.** PCR mapping of the borders of Chr XVI-L erosion (**A**) and Chr X-R erosion (**B**) in SY14 *tlc1*Δ survivors.
**Figure supplement 2.** Determination of rTG Type survivors by PCR (**A**) and fusion junction sequences of non-TG Type survivors (**B**) of SY14 *tlc1*Δ mutants.
**Figure supplement 3.** Fusion junctions of rTG Type in SY14 *tlc1*Δ survivors.
**Figure supplement 4.** Telomerase null cells may experience inter- and/or intra-chromosome fusions to survive with the decrease of chromosome numbers in budding yeast.

---

multiple Y'-elements exist in either subtelomeric regions in different chromosomes or Y'-elements containing extrachromosomal circles (*Larrivée and Wellinger, 2006*; *Lin et al., 2005*), which facilitates telomere recombination. There are no Y'-elements or counterparts thereof in fission yeast. Alternatively, or additionally, the survivors with chromosome fusions presented less of a growth advantage (e.g., slower growth rates) than canonical survivors and were outcompeted during continuous culture.

## The Rad52 pathway plays a dominant role in chromosomal end-to-end fusion in SY14 *tlc1*Δ cells

To investigate whether Rad52 contributes to chromosomal circularization in SY14 *tlc1*Δ cells, we constructed the SY14 *tlc1*Δ *rad52*Δ pRS316-*TLC1* strain. The plasmid-borne wild-type *TLC1* gene (pRS316-*TLC1*) was counterselected on 5'-FOA plates. The frequency of survivor emergence in the SY14 *tlc1*Δ *rad52*Δ and SY14 *tlc1*Δ strains was measured via the cell viability assay (see Materials and methods). The results showed that the double deletion of *TLC1* and *RAD52* in the SY14 strain accelerated senescence (*Figure 7A*), and SY14 *tlc1*Δ *rad52*Δ survivors eventually appeared but seemed to be much more difficult to obtain than SY14 *tlc1*Δ survivors (*Figure 7A*), suggesting that Rad52 plays an important role in survivor generation. Southern blotting assay revealed that telomere signals of ~1.3 kb were not detected in SY14 *tlc1*Δ *rad52*Δ survivors (*Figure 7B*), indicating chromosomal circularization. Further chromosome fusion mapping and sequencing results showed that no rTG Type survivors were generated in SY14 *tlc1*Δ *rad52*Δ cells (*Figure 7C*; *Figure 7—figure supplements 1–3*). The SY14 *tlc1*Δ *yku70*Δ survivor types were also examined (*Figure 7—figure supplement 4*), and the rTG Type survivor generation rate was 81%, comparable to 84% in SY14 *tlc1*Δ cells (*Figure 7C* and *Figure 7—figure supplement 4C*). These data indicate that the *RAD52* pathway plays a dominant role in chromosomal end-to-end fusion in SY14 *tlc1*Δ cells. We then examined whether Pol32 played a role in chromosome circularization in SY14 *tlc1*Δ cells. Thus, we deleted *POL32* in SY14 *tlc1*Δ pRS316-*TLC1* cells, and the plasmid-borne wild-type gene was counterselected on 5'-FOA plates thereafter. The mutants were passaged on plates ten times until survivors were generated. Both SY14 *tlc1*Δ and SY14 *tlc1*Δ *pol32*Δ cells senesced at the 3rd re-streak on plates, and survivors gradually emerged at the 7th re-streak (*Figure 7D*), suggesting that the deletion of *POL32* in the SY14 *tlc1*Δ strain did not affect the development of survivors. Further chromosome fusion mapping (*Figure 7—figure supplement 5A and B*) and fusion type assay (*Figure 7—figure supplement 5C*) revealed that the rTG Type survivor rate in SY14 *tlc1*Δ *pol32*Δ cells was 100%, which was higher than the 84% rate observed in SY14 *tlc1*Δ cells (*Figure 7C*). These data suggest that Pol32 is not required for the generation of rTG Type survivors but might function in non-TG Type development in SY14 *tlc1*Δ cells. Notably, Pol32 appeared to affect the chromosome fusion rate in *cdc13*Δ cells (*Figure 4E*). We speculate that this discrepancy could likely be attributed to different telomere structures between SY14 *tlc1*Δ and *cdc13*Δ cells.

## Discussion

The haploid yeast cells of *Saccharomyces cerevisiae* used in laboratories (such as BY4742) usually contain 16 chromosomes and 32 telomeres. The physical and functional interactions between

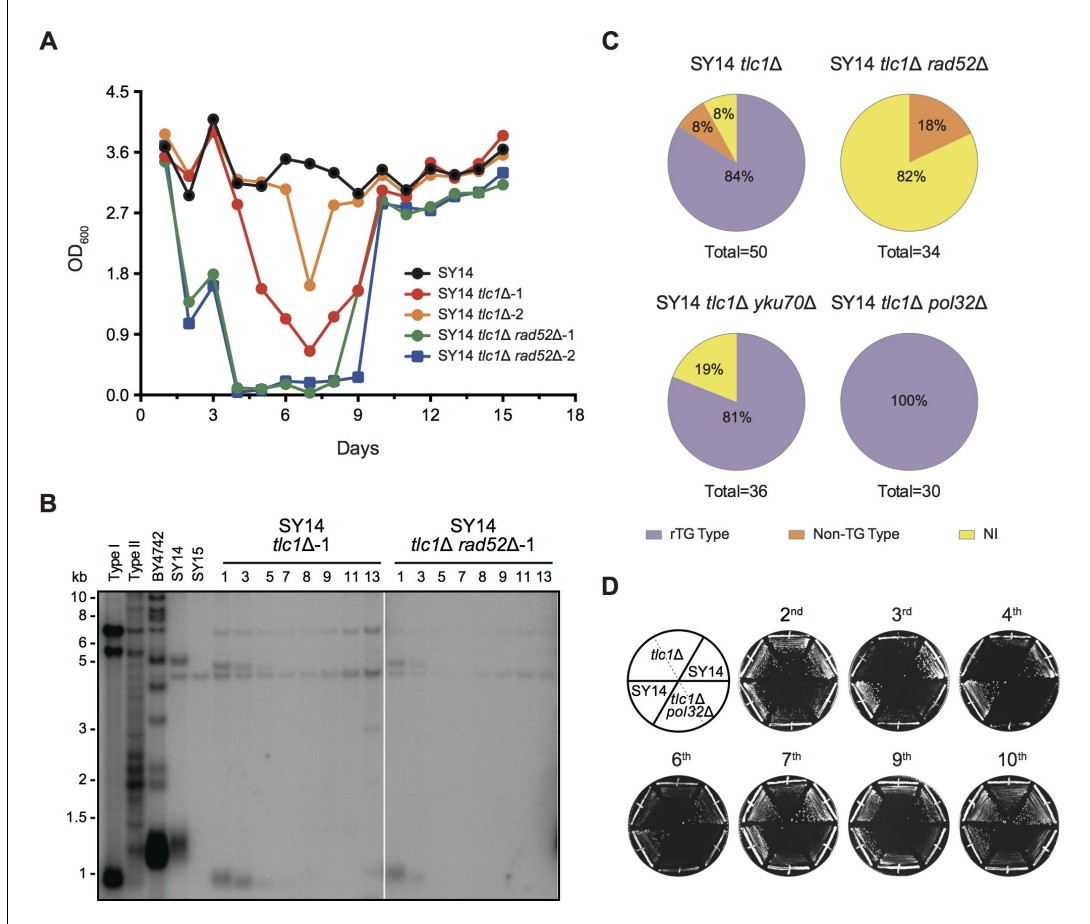

**Figure 7.** Survivors in SY14 *tlc1Δ* have a circular chromosome. (**A**) Senescence assay in liquid medium. The growth (OD$_{600}$) of SY14 (black), SY14 *tlc1Δ* (red, orange) and SY14 *tlc1Δ rad52Δ* (green, blue) strains were monitored every 24 hr for 15 days. (**B**) Telomere Southern blotting analysis of SY14 *tlc1Δ* and SY14 *tlc1Δ rad52Δ* survivors. Genomic DNA of the SY14 *tlc1Δ* and SY14 *tlc1Δ rad52Δ* strains assayed in (A) were digested by XhoI and subjected to a Southern blotting analysis. (**C**) Percentage of rTG Type, non-TG Type and not-identified (NI) survivors in SY14 *tlc1Δ* (n = 50), SY14 *tlc1Δ rad52Δ* cells (n = 34), SY14 *tlc1Δ yku70Δ* (n = 36) and SY14 *tlc1Δ pol32Δ* (n = 30). (**D**) Senescence assay of the SY14 *tlc1Δ* and SY14 *tlc1Δ pol32Δ* strains on solid medium. After eviction of the pRS316-*TLC1* plasmid in SY14 *tlc1Δ TLC1* or SY14 *tlc1Δ pol32Δ TLC1* strains by 5'-FOA selection, two independent SY14 *tlc1Δ* and SY14 *tlc1Δ pol32Δ* clones were re-streaked ten times to allow survivors to form. SY14 was a control.

The online version of this article includes the following source data and figure supplement(s) for figure 7:

**Source data 1.** Senescence assay of SY14 tlc1Δ and SY14 tlc1Δ rad52Δ cells.
**Figure supplement 1.** PCR mapping of the borders of Chr XVI-L erosion in SY14 tlc1Δ rad52Δ survivors.
**Figure supplement 2.** PCR mapping of the borders of Chr X-R erosion in SY14 tlc1Δ rad52Δ survivors.
**Figure supplement 3.** Fusion junction sequences of non-TG Type survivors derived from SY14 tlc1Δ rad52Δ mutants.
**Figure supplement 4.** Borders of erosion (A and B) and rTG Type (C) of SY14 tlc1Δ yku70Δ survivors are defined by mapping and PCR amplification.
**Figure supplement 5.** Chromosomal circularization in SY14 tlc1Δ cells is independent of Pol32.

telomeres seem to complicate the study of telomere structure and function. Recently, we successfully constructed single-chromosome yeast strains, SY14 and SY15, which contain two or no telomeres, respectively (*Shao et al., 2019a*; *Shao et al., 2018*). In this work, we have taken advantage of these single-chromosome yeast strains to elucidate telomere-associated processes, some of which might be difficult to observe in yeast with multiple chromosomes.

Cdc13, Stn1 and Ten1 form a complex (CST) to protect telomeres and regulate telomere replication (*Chandra et al., 2001*; *Churikov et al., 2013*; *Grossi et al., 2004*; *Nugent et al., 1996*; *Pennock et al., 2001*). The deletion of any of the CST genes results in an immediate cessation of cell division, likely due to telomere deprotection (*Garvik et al., 1995*; *Grandin et al., 2001b*; *Grandin et al., 1997*). Accordingly, *cdc13-1*, *stn1-13* or *ten1-31* temperature-sensitive mutants all

display long single-strand telomeric $TG_{1-3}$ DNA at restrictive temperatures (*Garvik et al., 1995*; *Grandin et al., 2001b*; *Grandin et al., 1997*). These studies suggest that *CDC13*, *STN1* and *TEN1* function epistatically in telomere protection. This notion is strengthened but also challenged by the results obtained from our single-chromosome yeast. In SY15 cells containing a single circular chromosome, the deletion of *CDC13*, *STN1* or *TEN1* does not appear to affect cell growth or viability (*Figure 2B and C*), indicating that the essentiality of CST is mainly due to telomere protection. In contrast, *RAP1* is still essential for the viability of SY15 cells (*Figure 2A*), indicating the importance of the transcriptional function of Rap1 (*Alexander and Zakian, 2003*; *Lieb et al., 2001*).

In single-chromosome SY14 cells, the deletion of *CDC13* results in telomere deprotection, leading to inviability of most of the cells. Unexpectedly, approximately 1‰ of these cells survived through Rad52-mediated chromosomal end-to-end fusion (*Figure 4C*). Although this phenomenon appears to be common in fission yeast in the absence of telomerase or telomere binding proteins (*Baumann and Cech, 2001*; *Martín et al., 2007*; *Nakamura et al., 1998*; *Tashiro et al., 2017*; *Wang and Baumann, 2008*), it has not been clearly observed or characterized in budding yeast cells with multiple chromosomes. More interestingly, chromosomal fusion events were ~30 fold less common in SY14 *stn1Δ* or SY14 *ten1Δ* cells than in SY14 *cdc13Δ* cells (*Figure 5A*), and the survivor formation rate was significantly improved by the further deletion of *CDC13* in the *stn1Δ* and *ten1Δ* strains (*Figure 5C*), reinforcing the conclusion that it is Cdc13, and probably not Stn1 and/or Ten1, that plays the major inhibitory role in chromosomal fusion (*Grandin and Charbonneau, 2003*; *Grandin et al., 2001a*). This idea is supported by the biochemical finding that Cdc13 binds to telomeric DNA with an exceptionally high affinity (*Nugent et al., 1996*) and thereby inhibits the HR pathway. On the other hand, the chromosome fusion frequency in SY14 *cdc13Δ stn1Δ* or SY14 *cdc13Δ ten1Δ* cells is >3.8 fold lower than that in SY14 *cdc13Δ* cells (*Figure 5C*), suggesting that Stn1 and Ten1 play roles in telomere protection independent of Cdc13 and limit telomere accessibility to prevent inappropriate degradation. This hypothesis is supported by the biochemical findings that both Stn1 and Ten1 exhibit weak telomeric DNA-binding activity in vitro (*Gao et al., 2007*; *Qian et al., 2010*; *Qian et al., 2009*), and the genetic results showing that the fusion of the DNA-binding domain of Cdc13 to Stn1 or the overexpression of Stn1/Ten1 bypasses the requirement for Cdc13 (*Pennock et al., 2001*; *Petreaca et al., 2006*).

We were relatively fortunate to detect rTG Type chromosomal fusions in SY14 *cstΔ* or SY14 *tlc1Δ* cells. Presumably, the two telomeres in SY14 *cstΔ* or SY14 *tlc1Δ* cells exhibit an equal probability of undergoing degradation, but the $5'$-$TG_{1-3}$-$3'$/$5'$-$C_{1-3}$A-$3'$ sequences retained in the circular chromosomes of the recovered clones all came from Chr XVI-L (*Figure 4A*). The likely reasons for this are as follows: (1) there is a 34 bp $5'$-$(CA)_{17}$-$3'$/$5'$-$(TG)_{17}$-$3'$ repeat sequence at the 43.9 kb locus proximal to the Chr X-R telomere, whose $5'$-$(CA)_{17}$-$3'$ sequence matches well with the $5'$-$TG_{1-3}$-$3'$ sequence in the Chr XVI-L telomere; and (2) there is no $5'$-$C_{1-3}$A-$3'$/$5'$-$T_{1-3}$G-$3'$ repeat sequence in the Chr XVI-L subtelomere region, which is located telomere-proximal to the essential gene *PLC1*. Accordingly, the 255 bp $C_{1-3}$A/$TG_{1-3}$ telomere sequence, which was inserted more proximally to the telomeres than the $(CA)_{17}$ repeat sequence, was preferentially utilized as a fusion sequence (*Figure 5—figure supplement 6*). We are able to recover chromosomal fusion clones of the 'non-TG Type'. However, their emergence rate was much lower than that of 'rTG Type' clones (*Figures 4D*, *5B, D* and *7C*), probably because telomere-proximal non-TG sequences showing microhomology are rare, or the efficiency of microhomology-mediated chromosomal fusions is quite low. Thus, the fortuity of identifying a single-chromosome yeast with the original Chr X-R telomere, which contains a 34 bp $5'$-$(CA)_{17}$-$3'$/$5'$-$(TG)_{17}$-$3'$ repeat sequence, was beyond our expectations because the availability of a reasonable number of 'rTG Type' survivors greatly facilitates the quantification of chromosomal fusions. Chromosomal fusion is readily detected in single-chromosome SY14 yeast but not in sixteen-chromosome BY4742 yeast (wild type) lacking Cdc13 or telomerase. Even if chromosome fusion takes place in BY4742 *cdc13Δ* or BY4742 *tlc1Δ* cells, there is little or no chance to recover such survivors, because the low efficiency of chromosome fusion, interchromosomal telomere interaction and unforeseeable growth defect make the mission impossible.

Both rTG Type (TG/CA) and non-TG Type (other microhomologous sequences) chromosomal fusion appear to require Rad52 (*Figure 4C*), a critical factor for almost all kinds of homologous DNA sequence-mediated activities (*San Filippo et al., 2008*; *Symington, 2002*), such as the double-strand break repair (DSBR) (*Szostak et al., 1983*), synthesis-dependent strand annealing (SDSA) (*Nassif et al., 1994*), single-strand annealing (SSA) (*Fishman-Lobell et al., 1992*; *Lin et al., 1984*;

*Maryon and Carroll, 1991*) and break-induced replication (BIR) (*McEachern and Haber, 2006*; *Morrow et al., 1997*) pathways. Single-strand annealing is a highly mutagenic homologous recombination pathway that proceeds via the annealing of repeated sequences flanking DSB sites (*Bhargava et al., 2016*; *Davis and Symington, 2001*; *Fishman-Lobell et al., 1992*; *Ivanov et al., 1996*; *Lin et al., 1984*; *Lin et al., 1990*; *Maryon and Carroll, 1991*; *Symington, 2002*). The process is initiated by the resection of the 5′-ends and exposure of homologous regions oriented in the same direction. After annealing to each other, the nonhomologous tails are removed. This is followed by DNA synthesis that fills in the gaps and ligation. Given that telomere deprotection in SY14 *cstΔ* and SY14 *tlc1Δ* cells involves end resection and homologous sequence pairing (*Figure 4A and B*; *Figure 3—figure supplements 2–4*; *Figure 4—figure supplement 3*; *Figure 5—figure supplements 3 and 5*; *Figure 6—figure supplements 2 and 3*), we favor the model in which rTG Type and non-TG Type chromosomal fusions are mainly achieved through the SSA pathway. However, we cannot rule out the possibility that chromosomal circularization occurs via the MMEJ or BIR pathway. The latter model is supported by the observation that chromosome fusion in *cdc13Δ* cells is partially dependent on Pol32 (*Figure 4E*), which is involved in both MMEJ and BIR pathways (*Lee and Lee, 2007*; *Lydeard et al., 2007*).

# Materials and methods

## Yeast strains and plasmids

Information about yeast strains used in this study is listed in *Supplementary file 2*. The plasmids for gene deletion were constructed as described previously (*Sikorski and Hieter, 1989*). In brief, two DNA segments flanking the target gene were amplified by PCR, and then the products were digested with appropriate restriction enzymes and inserted into pRS plasmids in tandem order. CEN plasmids were constructed by cloning DNA sequences including endogenous promoter, terminator and open reading frame of target genes into pRS316 plasmid. Plasmids were introduced into budding yeast by standard procedures and genetic cross (mating and tetrad dissection), and transformants were selected on auxotrophic medium.

## Successive passages assay

Several clones of indicated strains were picked and re-streaked on extract-peptone-dextrose (YPD) plates and grown until the emergence of single colonies (25 cell divisions) at 30℃. This procedure was repeated dozens of times at intervals of two days. For the strains which display a slightly reduced growth rate, the intervals were extended to three days to ensure similar population doublings.

## Cell growth assay

Three individual colonies of indicated strains were inoculated into 5 ml YPD medium and incubated at 30℃ to saturation. The cell cultures were diluted in 30 ml of fresh YPD medium to the density at $OD_{600} = 0.1$. Then the density of cells was measured by spectrometry ($OD_{600}$) hourly.

## Pulsed-field gel electrophoresis (PFGE) analysis

Yeast chromosome samples embedded in agarose plugs were prepared according to the manufacturer's instructions (Bio-Rad), with the following modifications. For each plug, $1 \times 10^8$ cells were washed with 50 mM EDTA (pH 8.0) and then washed with buffer containing 20 mM Tris (pH 7.2), 40 mM NaCl and 200 mM EDTA (pH 8.0). The cells were then resuspended in buffer containing 20 mM Tris (pH 7.2), 40 mM NaCl, 200 mM EDTA (pH 8.0), 2 mg/ml lyticase (Sigma) and 10 mg/ml lysing enzyme (Sigma). The suspension was mixed with one volume of 1.5% low melt agarose (Bio-Rad) at 50℃ and allowed to solidify in the plug mold (Bio-Rad) at 4℃ for 30 min. The agarose plugs were released from the molds and incubated with buffer containing 10 mM Tris (pH 7.2), 100 mM EDTA and 1 mg/ml lyticase at 37℃ for 5 hr without agitation. The buffer was removed and the blocks were rinsed with buffer containing 20 mM Tris (pH 8.0) and 50 mM EDTA. The blocks were then incubated with buffer containing 100 mM EDTA (pH 8.0), 0.2% sodium deoxycholate, 1% sodium lauryl sarcosine and 1 mg/ml Proteinase K for 48 hr at 50℃ without agitation. Plugs were washed for 1 hr four times in 20 mM Tris (pH 8.0) and 50 mM EDTA at room temperature with gentle agitation. The

Agarose-embedded DNA plugs were stored at 4°C in 20 mM Tris (pH 8.0) and 50 mM EDTA. Pulsed-field gel electrophoresis was carried out using a CHEF-DR II Pulsed Field Electrophoresis Systems (Bio-Rad) under the following conditions: 0.8% agarose gel in 1 × TAE; temperature, 14°C; first run: initial switch time, 1200 s; final switch time, 1200 s; run time 24 hr; voltage gradient, 2 V/cm; angle, 96°; second run: initial switch time, 1500 s; final switch time, 1500 s; run time 24 hr; voltage gradient, 2 V/cm; angle, 100°; third run: initial switch time, 1800 s; final switch time, 1800 s; run time 24 hr; voltage gradient, 2 V/cm; angle, 106°. After electrophoresis, the gel was stained with ethidium bromide and photographed.

## Telomere southern blotting

Telomere Southern blotting assay was performed as previously described (*Hu et al., 2013*). In short, genomic DNA was extracted from indicated yeast strains by a phenol chloroform method, digested with XhoI, separated on 1% agarose gel, transferred to Hybond-N+membrane (GE Healthcare) and then hybridized with telomere-specific probe.

## Serial dilution assay

A single colony of indicated yeast strains was inoculated into 3 ml selective medium at 30°C to reach saturation. Then the cell cultures were adjusted to $OD_{600}$ of 0.5 and five-fold serially diluted, the equal amounts of the cell cultures (0.6 µl for 5′-FOA plates and 0.35 µl for Ura⁻) were spotted on 5′-FOA and selective medium (Ura⁻) plates. Plates were incubated at 30°C for four days before photography.

## Molecular analysis of circular chromosomes

Fusion points were determined by PCR and DNA sequencing. Genomic DNA was extracted from indicated yeast strains by a phenol chloroform method and resuspended in ddH$_2$O. To determine how much sequences of each chromosome arm had lost before circularization, primers pairs located at different sites of each chromosome arm were used (listed in *Supplementary file 1*). PCR reactions (10 µl) contained 50 ng genomic DNA, 10 × Ex Taq Buffer (Mg$^{2+}$ Plus, 20 mM), 2 mM dNTPs, 4 µM of each primer and 0.5 U TaKaRa Ex Taq. The conditions were as follows: 94°C 3 min, then 25–30 cycles of 94°C 30 s, 55°C 30 s and 72°C 45 s, followed by 72°C 10 min.

To amplify fusion points, reactions (50 µl) contained 100 ng genomic DNA, 10 × LA Taq Buffer II (Mg$^{2+}$ Plus), 10 mM dNTPs, 10 µM of each primer and 1.25 U TaKaRa LA Taq. The conditions were as follows: 94°C 3 min, then 25–30 cycles of 94°C 30 s, 55°C 30 s and 72°C 2.5 min, followed by 72°C 10 min. PCR products were purified, and then either sequenced directly or cloned into the pMD18 T Vector (TaKaRa) for sequencing.

## Quantitative survivor formation assay

A single colony of indicated yeast strains was inoculated into 5 ml selective medium (Ura⁻) at 30°C to reach saturation. Then the cell cultures were adjusted to $OD_{600}$ of 0.5 and five-fold serially diluted. The equal amounts of the cell cultures (100 µl) at indicated diluted concentration were plating on either 5′-FOA or selective medium (Ura⁻) plates. To ensure similar population doublings, selective medium (Ura⁻) plates and 5′-FOA plates were respectively incubated at 30°C for two or three days before counting. Survivor formation rate was determined by the ratio of colonies recovered from 5′-FOA plates to the colonies recovered from selective medium (Ura⁻) plates. Over 30 single colonies were randomly picked, then genomic DNA was extracted and fusion points were mapped and determined by PCR-sequencing. According to the fusion sequences, the survivor types were categorized as rTG Type, non-TG Type and NI (not identified).

## Plasmid repair assay

The plasmid repair assay was performed as previously reported (*Boulton and Jackson, 1996*; *Wilson and Lieber, 1999*; *Zhang and Paull, 2005*). In brief, the test plasmid pRS316 contains a CEN/ARS cassette for plasmid maintenance and a *URA3* marker for selection. In vitro, the plasmid substrate was linearized with EcoRI, generating two ends which are not homologous to yeast genomic sequences, and then the digested plasmids were transformed into SY14, SY14 *lig4Δ* and SY14 *yku70Δ* strains. Yeast transformants bearing only the recircularized plasmids survived on the selective

medium. Parallelly, the yeast cells transformed with an equal amount of undigested plasmid pRS316 were plated to uracil- YC medium. Colonies were counted after the plates had been incubated for two days and the efficiency of NHEJ was measured by dividing the number of cells with the cut plasmid by that with the uncut plasmid.

## Cell viability assay

Three individual colonies of indicated strains were inoculated into 5 ml YPD medium and incubated at 30℃ to saturation. Every 24 hr, cell densities were measured and then the cell cultures were diluted to the density at $OD_{600}$ of 0.01 with fresh YPD medium. This procedure was repeated several times until the cell densities remain relatively constant. Besides, the samples at indicated time points were collected for telomere length analysis.

## Single-colony streaking assay

A single colony of indicated yeast strains was re-streaked on YPD plates at 30℃. Normally, This procedure was repeated six to ten times every 48 hr (25 cell divisions) to allow the generation of survivors.

## Acknowledgements

We thank members of Zhou lab and Qin lab for discussions and suggestions for this project. This project was supported by grants from the National Key Research and Development Program of China (2016YFA0500701), National Natural Science Foundation of China (NSFC) (31230040), Shanghai Research Project (18JC1420202) and the Strategic Priority Research Program of the Chinese Academy of Sciences (XDB19000000).

## Additional information

### Funding

| Funder | Grant reference number | Author |
|---|---|---|
| The National Key Research and Development Program of China | 2016YFA0500701 | Zhi-Jing Wu<br>Jia-Cheng Liu<br>Xin Man<br>Xin Gu<br>Ting-Yi Li<br>Chen Cai<br>Ming-Hong He<br>Jin-Qiu Zhou |
| National Natural Science Foundation of China | 31230040 | Zhi-Jing Wu<br>Jia-Cheng Liu<br>Xin Man<br>Xin Gu<br>Ting-Yi Li<br>Chen Cai<br>Ming-Hong He<br>Jin-Qiu Zhou |
| Shanghai Research Project | 18JC1420202 | Zhi-Jing Wu<br>Jia-Cheng Liu<br>Xin Man<br>Xin Gu<br>Ting-Yi Li<br>Chen Cai<br>Ming-Hong He<br>Yangyang Shao<br>Ning Lu<br>Xiaoli Xue<br>Zhongjun Qin<br>Jin-Qiu Zhou |

| Chinese Academy of Sciences | Strategic Priority Research Program XDB19000000 | Zhi-Jing Wu<br>Jia-Cheng Liu<br>Xin Man<br>Xin Gu<br>Ting-Yi Li<br>Chen Cai<br>Ming-Hong He<br>Zhongjun Qin<br>Jin-Qiu Zhou |

The funders had no role in study design, data collection and interpretation, or the decision to submit the work for publication.

### Author contributions

Zhi-Jing Wu, Conceptualization, Data curation, Formal analysis, Investigation, Visualization, Methodology, Writing - original draft; Jia-Cheng Liu, Investigation, Writing - review and editing, Participated in discussion; Xin Man, Xin Gu, Ting-Yi Li, Investigation, Participated in discussion; Chen Cai, Ming-Hong He, Resources, Writing - review and editing, Participated in discussion; Yangyang Shao, Ning Lu, Xiaoli Xue, Zhongjun Qin, Resources, Methodology, Participated in discussion; Jin-Qiu Zhou, Conceptualization, Supervision, Funding acquisition, Project administration, Writing - review and editing, Participated in discussion

### Author ORCIDs

Zhi-Jing Wu (iD) https://orcid.org/0000-0002-9203-0252
Jin-Qiu Zhou (iD) https://orcid.org/0000-0003-1986-8611

### Decision letter and Author response

Decision letter https://doi.org/10.7554/eLife.53144.sa1
Author response https://doi.org/10.7554/eLife.53144.sa2

## Additional files

### Supplementary files

- Supplementary file 1. Mapping primers used in this study.
- Supplementary file 2. Yeast strains used in this study.
- Transparent reporting form

### Data availability

All data generated or analysed during this study are included in the manuscript and supporting files.

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
