## [Decision Letter]

**Acceptance summary:**

The authors use a recently created elegant system in which all 16 yeast chromosomes are fused into one chromosome and then compare cellular phenotypes and telomere behavior in cells that live with this one linear chromosome to those of cells that live with the one chromosome circularized. The results show that the telomeric ssDNA binding protein Cdc13 is the critical component for fusion prevention, while the Stn1 and Ten1 protein do contribute in different ways. It also appears that the replication associated function of the CST heterotrimer is not as critical as previously anticipated. These studies thus provide an unprecedented opportunity to detail the functions of essential chromosomal capping genes in telomeric repeat maintenance in the presence or absence of telomerase.

**Decision letter after peer review:**

Thank you for submitting your article "Cdc13 but not Stn1 or Ten1 plays a predominant role in preventing chromosomes from end-to-end fusion" for consideration by *eLife*. Your article has been reviewed by three peer reviewers, one of whom is a member of our Board of Reviewing Editor, and the evaluation has been overseen by Jessica Tyler as the Senior Editor. The following individual involved in review of your submission has agreed to reveal their identity: Connie Nugent (Reviewer #3).

The reviewers have discussed the reviews with one another and the Reviewing Editor has drafted this decision to help you prepare a revised submission.

The reviewers felt that the single chromosome yeast strains constitute a superb system to assess telomeric functions and phenotypes, including the precise functions of associated proteins. To be able to manipulate strains in complete absence of telomeres allows for very clear and unencumbered experiments and this therefore is a great plus for the paper's experimental lines. However, all three reviewers also thought that as is, there was a bit of a lack of depth and new insights into telomere biology. For example, there is already indirect evidence that CST proteins can be non-essential. On the other hand, given that there is much talk about the interactions between passing replication forks and the end-activities, or capping, of certain factors, they thought that certain rather simple experiments could bring this manuscript up to the level we expect for *eLife*. After thorough discussion, the reviewers request the following experiments:

1) A major debate in the field is the question of whether the CST-complex does function alike RPA but on telomeric repeat DNA. It is important to recognize that this would be a telomere replication function, not necessarily a protection function. Hence, the conclusion in subsection “The essentiality of CST is solely attributed to its roles in telomere protection” that CST is solely required for telomere protection is not (yet) supported by the experiments. However, this could easily be tested by inserting an adequate telomeric repeat tract (250 bp or so) in both orientations somewhere on the single linear and circular chromosome strains. Then let those four strains lose CST components and monitor the stability of the inserted GT tracts. If there really is no significant internal function for any of the CST components (*cdc13*Δ ; *stn1*Δ *ten1*Δ; or *cdc13/stn1* double; *cdc13/ten1* and *stn1/ten1*), then these non-terminal tracts should be stable in either orientation on the circular chromosome.

2) All of the yeast survivors require that Pol32 gene be present. If the circularization you observe after *Tlc1* (telomerase) loss is independent of those mechanisms, this circularization should be independent of Pol32. So, verify the *tlc1*Δ *cst*Δ with a *pol32*Δ.

3) Given your results and those in *S. pombe* with three chromosomes, the question arises as to when are there too many linear chromosomes for circularization as a solution to telomere problems? You should be able to readily address that by deleting Tlc1 or Est2 from your intermediate linear chromosome strains. I.e. you show here a single linear chromosome does circularize; how about the intermediate strain with three chromosomes (like *S. pombe*?), with five chromosomes, with 10 chromosomes? The results from this experiment would certainly broaden the scope of the paper enormously.

4) For technical reasons, it would be good to conclusively assess that NHEJ is indeed still functional in the single linear chromosome strain. A simple linear plasmid transformation and circularization assay would be enough and use *lig4*Δ as negative control.

The addition of the above experiments was deemed essential by all reviewers for the paper to go forward. Below I add some other comments that were noted by reviewers and that you may or may not address with experiments. However, the reviewers expect them to be at least commented on in the revised manuscript.

5) SY14 *tlc1* cells carry all the CDC13, STN1 and TEN1 genes. If Cdc13 plays a key role in preventing telomeric fusion, SY14 *tlc1* survivors are expected to contain non-TG type fusion more frequently compared with rTG fusion. However, SY14 *tlc1* and SY14 *cdc13* cells have a similar profile in rTG and nonTG fusion (Figure 4, 5 and 7). Some better explanation is required.

6) In the PCR mediated breakpoint mappings (first occurrence Figure 3—figure supplement 1A and B). It is my understanding that a black dot means that the PCR was productive and this site present in the clone. Therefore, please insert a black dot into ALL squares after the first occurrence of a black dot, indicating that from here on inwards, all PCRs were positive and the chromosomal loci present (not rearranged). As is, those squares are empty, as if the PCR did not work; yet the bands are there.

7) We found it remarkable that in the Rad52+ cells, the fusion points almost invariably retained all of the Y'-telomere (including some repeats) but had lost over 30 kb of sequences from the other end. The SSA and related mechanisms predict an extended ssDNA for the annealing and such an unbalanced resection is somewhat surprising. Could they comment on what happens to the Y' TELOXVIL in the probably significantly long time until resection hits that hot spot in the subtelomere of TELOXR? Is there a difference in the replication timing of the two ends?

8) To what do you attribute the telomere length fluctuation observed in Figure 1F among the streakout colonies?

9) In Discussion, one could bring in the parallel example of chromosomal circularization observed in *S. pombe*, where having just a few chromosomes allows circular survivors to arise when telomerase is deficient. (Paragraph three) I think mentioning this fission yeast case would benefit the discussion because it is an example of cells using chromosome circularization to survive absence of telomerase.

[Editors' note: further revisions were suggested prior to acceptance, as described below.]

Thank you for submitting the revised version of your article "Cdc13 but not Stn1 or Ten1 plays a predominant role in preventing chromosomes from end-to-end fusion" for consideration by *eLife*. Your article has been re-reviewed by three peer reviewers, including Raymund Wellinger as the Reviewing Editor and Reviewer #1, and the evaluation has been overseen by Jessica Tyler as the Senior Editor.

The reviewers have discussed the reviews with one another and the Reviewing Editor has drafted this decision to help you prepare a revised submission.

Reviewers all agreed that the additional data generated for the revised manuscript is interesting and renders the paper more suitable for a general readership. However, certain additions and explanations appear to lack specificity and precision and they also thought that the survivor data must be incorporated, given the rather interesting findings. The reviewers agreed that the requested data be incorporated in particular if you do have the data in hand.

Specifically:

1) Please do include the data on the multiple and single chromosome yeasts and their mode of survival in the absence of telomerase. While a full mechanistic understanding may not be achieved here, it does provide for essential insights into budding yeast chromosome biology. It would also validate the statement that the observations "could never be seen in multiple chromosome yeast".

2) The GT-tract stability data is considered extremely interesting, albeit a bit incomplete. Could growth characteristics, viability platings or other data you have address the issue of how frequent complete losses of these tracts occur and what this may cause? Is it possible that such major replication problems go undetected in your assay (because of inviability of the cells) and therefore skew the interpretation derived from it? In case there is no experimental data addressing this possibility directly, you would have to tone down the Discussion "the essentiality of CST is solely attributed to telomere protection" and include the caveat that one-step complete losses may not be recovered in your system

3) Role of Pol32 in telomere fusion (Figure 4E): Pol32 appears to affect the chromosome fusion rate in *cdc13*Δ cells. This discrepancy was likely attributed to different telomere structures between SY14 *tlc1Δ* and *cdc13*Δ cells. The reviewers thought that this speculative idea would need some more direct evidence. Otherwise, it would be better to say "we speculate different telomere structures between SY14 *tlc1Δ* and *cdc13*Δ cells".

---

## [Author Response]

After thorough discussion, the reviewers request the following experiments:1) A major debate in the field is the question of whether the CST-complex does function alike RPA but on telomeric repeat DNA. It is important to recognize that this would be a telomere replication function, not necessarily a protection function. Hence, the conclusion in subsection “The essentiality of CST is solely attributed to its roles in telomere protection” that CST is solely required for telomere protection is not (yet) supported by the experiments. However, this could easily be tested by inserting an adequate telomeric repeat tract (250 bp or so) in both orientations somewhere on the single linear and circular chromosome strains. Then let those four strains lose CST components and monitor the stability of the inserted GT tracts. If there really is no significant internal function for any of the CST components (cdc13Δ ; stn1Δ ten1Δ; or cdc13/stn1 double; cdc13/ten1 and stn1/ten1), then these non-terminal tracts should be stable in either orientation on the circular chromosome.

We appreciate the constructive suggestion. According to this suggestion, we used several single circular chromosome yeast strains, including SY15 *cdc13*Δ *CDC13*, SY15 *stn1*Δ *STN1*, SY15 *ten1*Δ *TEN1*, SY15 *cdc13*Δ *stn1*Δ *CDC13 STN1*, SY15 *cdc13*Δ *ten1*Δ *CDC13 TEN1* and SY15 *stn1*Δ *ten1*Δ *STN1 TEN1* (SY15 *cst*Δ *CST*), to address this issue. Briefly, we inserted a 255 bp-long telomeric sequence into a genomic locus between genes *PGU1* and *YJR154W* in both orientations in SY15 *cst*Δ *CST* strains. The yielding strains, which were designated as SY15^CA255^ and SY15^TG255^, respectively, were passaged on plates for five times (~100 population doublings). The inserted TG255/CA255 sequences determined by Southern blotting were successfully transmitted from generation to generation in all the SY15^CA255^ and SY15^TG255^ strains, regardless of the presence or absence of CST. In order to find out whether there was any tiny contraction or expansion of TG255/CA255 sequences after ~100 rounds of replication, we performed PCR analysis, and the amplified DNA fragments from all of the strains were in the same size. Further sequencing results of three independent clones of each *cst* mutants confirmed that there were no mutations after ~100 population doublings. These results indicate that CST complex does not affect the replication of non-terminal telomeric sequences, and in other words, CST-complex likely has no other telomere replication function than recruitment of Polα for lagging strand synthesis. These results have been presented as Figure 2D and E; Figure 2—figure supplements 1 and 2 in the revised manuscript.

2) All of the yeast survivors require that Pol32 gene be present. If the circularization you observe after Tlc1 (telomerase) loss is independent of those mechanisms, this circularization should be independent of Pol32. So, verify the tlc1Δ cstΔ with a pol32Δ.

As the reviewer’s suggestion, we deleted *POL32* in SY14 *tlc1*Δ pRS316-*TLC1* cells and the plasmid-borne wild-type genes were counter-selected on 5′-fluoroorotic acid (5′-FOA) plates thereafter. The mutants were passaged on plates for ten times until survivors generated. Both of the SY14 *tlc1*Δ and SY14 *tlc1*Δ *pol32*Δ cells senesced at the 3rd re-streak on plates and survivors gradually emerged at the 7th re-streak, suggesting that deletion of *POL32* in SY14 *tlc1*Δ strain did not affect the formation of survivors. Further chromosome-fusion mapping and fusion type assay revealed that the rTG Type survivor rate in SY14 *tlc1*Δ *pol32*Δ cells was 100%, higher than 84% in SY14 *tlc1*Δ cells. These data suggest that Pol32 is not required for the generation of rTG Type survivors in SY14 *tlc1*Δ cells, but might function in non-TG Type formation. This result has been presented in the revised manuscript as Figure 7C and D, Figure 7—figure supplement 5.

In addition, we also constructed SY14 *pol32*Δ *cdc13*Δ pRS316-*CDC13* strain to examine whether chromosomal circularization relies on Pol32. The quantitative survivor formation assay showed that further deletion of *POL32* in *cdc13*∆ mutant resulted in an approximately 3-fold decrease of chromosome fusion rates compared to single deletion of *CDC13*, indicating chromosomal circularization in SY14 *cdc13*Δ survivors partially depends on Pol32. But Pol32 seems not to affect chromosome fusion in SY14 *tlc1*Δ cells, supporting the model that in SY14 *tlc1*Δ cells, the presence of Cdc13 at telomeres suppresses chromosome fusion. We included the result in our revised manuscript as Figure 4E.

3) Given your results and those in S. pombe with three chromosomes, the question arises as to when are there too many linear chromosomes for circularization as a solution to telomere problems? You should be able to readily address that by deleting Tlc1 or Est2 from your intermediate linear chromosome strains. I.e. you show here a single linear chromosome does circularize; how about the intermediate strain with three chromosomes (like S. pombe?), with five chromosomes, with 10 chromosomes? The results from this experiment would certainly broaden the scope of the paper enormously.

Thank this reviewer for the suggestion. Chromosome fusion takes place in fission yeast, which usually contains three chromosomes, in the absence of telomerase or telomere binding protein (Baumann and Cech, 2001, Martin et al., 2007, Nakamura et al., 1998, Tashiro et al., 2017, Wang and Baumann, 2008). Since SY14 *tlc1*Δ cells survived via chromosome fusion, we wanted to know whether chromosome fusions readily occurred in budding yeast with the decrease of chromosome numbers. We deleted *TLC1* in a series of chromosome-fused yeast strains, including SY1, SY3, SY5, SY7, SY8, SY9, SY10, SY11 and SY12, which were derived from BY4742 strain and contained different numbers of chromosomes (Figure 6—figure supplement 4A) (Shao et al., 2018). Fifteen single colonies of each *tlc1*Δ strains were immediately passaged on the plates for successive 7 to10 times after *TLC1* deletion. Because not every strain displayed typical senescence and growth crisis, we assumed that such a long time passages allow for the formation of survivors. The telomere pattern of these survivors was examined by the Southern blotting. The results showed that BY4742 *tlc1*∆ generated both Type I and Type II survivors (Lundblad and Blackburn, 1993, Teng et al., 2000, Teng and Zakian, 1999). With the reduction of chromosome numbers, the emergence frequency of type II survivors gradually decreased, while Y’ recombination was readily detected, an indication of an increase of type I survivors (Figure 6—figure supplement 4B-Ε). These data suggest that the efficiency of telomere recombination is affected by chromosomal numbers in budding yeast. Notably, some distinct bands (indicated by open arrows at the right of the panels) with a size larger or smaller than Y’-element occasionally emerged in SY7 *tlc1*∆ clones (eight chromosomes), as well as in SY *tlc1*∆ clones with fewer chromosomes, suggesting that in yeast strains with eight or fewer chromosomes, there might be chromosome fusions, however, most of the telomerase null cells likely prefer to utilize canonical telomere recombination pathway, instead of chromosome fusion, to survive. The reasons for that were not clear. One possibility was that multiple Y’-elements exist in either the sub-telomeric region in different chromosomes or Y’-element containing extra-chromosome circles (Lin et al., 2005), which facilitates telomere recombination. There is no Y’-element or its counterpart in fission yeast. Alternatively or additionally, the survivors with chromosome fusions had less growth advantage (e.g. slower growth rate) than canonical survivors, and were competed out during continuous culture. Because the mechanism of putative chromosome circularization and the reasons why the frequency of Type I (versus Type II survivors) increases with the reduction of chromosome numbers remain to be investigated further in the future, thus, we prefer not to include these results in the revised manuscript.

4) For technical reasons, it would be good to conclusively assess that NHEJ is indeed still functional in the single linear chromosome strain. A simple linear plasmid transformation and circularization assay would be enough and use lig4Δ as negative control.

Following the reviewers’ suggestion, we constructed SY14 *lig4*Δ and SY14 *yku70*Δ strains, and employed a plasmid repair assay as previously described (Boulton and Jackson, 1996, Wilson and Lieber, 1999, Zhang and Paull, 2005). The test plasmid pRS316 contains a CEN/ARS cassette for plasmid maintenance and a *URA3* marker for selection. The plasmid substrate was linearized with EcoRI, generating two ends which are not homologous to yeast genomic sequences, and then the digested plasmids were transformed into SY14, SY14 *lig4*Δ and SY14 *yku70*Δ strains. Yeast transformants bearing only the recircularized plasmids survived on the selective medium. Parallelly, the yeast cells transformed with an equal amount of undigested plasmid pRS316 were plated on uracil- YC medium. Colonies were counted and the efficiency of NHEJ was measured by dividing the number of cells with the cut plasmid by that with uncut plasmid. Compared with SY14 *lig4*Δ and SY14 *yku70*Δ strains, in which the NHEJ pathway is blocked, the NHEJ pathway in SY14 strain was functioning efficiently. The results were presented in Figure 4—figure supplement 4 in the revised manuscript.

The addition of the above experiments was deemed essential by all reviewers for the paper to go forward. Below I add some other comments that were noted by reviewers and that you may or may not address with experiments. However, the reviewers expect them to be at least commented on in the revised manuscript.5) SY14 tlc1 cells carry all the CDC13, STN1 and TEN1 genes. If Cdc13 plays a key role in preventing telomeric fusion, SY14 tlc1 survivors are expected to contain non-TG type fusion more frequently compared with rTG fusion. However, SY14 tlc1 and SY14 cdc13 cells have a similar profile in rTG and nonTG fusion (Figure 4, 5 and 7). Some better explanation is required.

The reviewers raised a point that has confused us as well. We could not provide a compelling explanation at this moment. We noticed that the length of the TG-sequence at fusion junctions in SY14 *tlc1*Δ cells was much shorter than that in SY14 *cdc13*Δ cells (Figure 3—figure supplements 2 and 3; Figure 6—figure supplement 3). This result suggests that chromosome fusion in SY14 *tlc1*Δ cells takes place after telomeres have experienced extensive shortening, while chromosome fusion in SY14 *cdc13*Δ cells seems to take place immediately after loss of Cdc13 protection. Thus, rTG fusion only tells the outcome of fusion, but doesn’t reflect any preceding event before fusion. We have included this notion in the revised manuscript.

6) In the PCR mediated breakpoint mappings (first occurrence Figure 3—figure supplement 1A and B). It is my understanding that a black dot means that the PCR was productive and this site present in the clone. Therefore, please insert a black dot into ALL squares after the first occurrence of a black dot, indicating that from here on inwards, all PCRs were positive and the chromosomal loci present (not rearranged). As is, those squares are empty, as if the PCR did not work; yet the bands are there.

We thank this reviewer for pointing out the mislabeling of our figures. We have inserted black dots into the corresponding squares in the revised manuscript as suggested. Besides, our calculation of “Distance from telomere X-R (kb)” was wrong and we correct it in the revised manuscript.

7) We found it remarkable that in the Rad52+ cells, the fusion points almost invariably retained all of the Y'-telomere (including some repeats) but had lost over 30 kb of sequences from the other end. The SSA and related mechanisms predict an extended ssDNA for the annealing and such an unbalanced resection is somewhat surprising. Could they comment on what happens to the Y' TELOXVIL in the probably significantly long time until resection hits that hot spot in the subtelomere of TELOXR? Is there a difference in the replication timing of the two ends?

We agreed with the logic the reviewers had pointed out. To address the reviewer’s concern, we constructed the SY14^CA255^ strain in which a 255 bp C_1-3_A sequence was inserted between genes *PGU1* and *YJR154W* on Chr X-R. The inserted C_1-3_A sequence is 20.9 kb away from telomeric TG_1-3_ sequence, more proximal than the (CA)_17_ sequence to the telomere of Chr X-R. Likewise, deletion of a CST component resulted in death of the majority of SY14^CA255^ cells, but survivors emerged. A single clone of each isogenic survivors was characterized, i.e. the erosion-points and fusion types of these survivors were determined by PCR-mapping. The results revealed that all of the clones were rTG Type, which utilized the CA255 instead of the “5′-(CA)_17_-3′” repeat sequence at their fusion junctions. Therefore, chromosome circularization preferentially occurred at the sites where complementary sequences existed on two telomeres. However, we could not exclude the possibility that different chromatin structures at two telomeres also contributed to the unbalanced erosion. These results have been presented in the revised manuscript as Figure 5—figure supplement 6.

8) To what do you attribute the telomere length fluctuation observed in Figure 1F among the streakout colonies?

Fluctuations of telomere length were indeed observed in different cell passages, as well as in different clones in the same passage, and the reason for the fluctuation was not clear. In *Saccharomyces cerevisiae*, the length of telomeres is maintained within a narrow size distribution but the telomeric DNA length at individual telomeres varies (Shampay and Blackburn, 1988, Zakian, 1996). When the chromosome numbers reduced from sixteen in wide-type BY4742 cells to one in SY14, the telomere length variation became apparent. Nevertheless, the telomeres in all of the clones varied within the range of ~250 bp to 350 bp, indicating that the telomere length fluctuation observed is the dynamic nature of the chromosome ends, which might be difficult to tell in wide-type cells in Southern blotting assay.

9) In Discussion, one could bring in the parallel example of chromosomal circularization observed in S. pombe, where having just a few chromosomes allows circular survivors to arise when telomerase is deficient. (Paragraph three) I think mentioning this fission yeast case would benefit the discussion because it is an example of cells using chromosome circularization to survive absence of telomerase.

Thanks for the comment. We added this part to the revised manuscript.

[Editors' note: further revisions were suggested prior to acceptance, as described below.]

Reviewers all agreed that the additional data generated for the revised manuscript is interesting and renders the paper more suitable for a general readership. However, certain additions and explanations appear to lack specificity and precision and they also thought that the survivor data must be incorporated, given the rather interesting findings. The reviewers agreed that the requested data be incorporated in particular if you do have the data in hand.Specifically:1) Please do include the data on the multiple and single chromosome yeasts and their mode of survival in the absence of telomerase. While a full mechanistic understanding may not be achieved here, it does provide for essential insights into budding yeast chromosome biology. It would also validate the statement that the observations "could never be seen in multiple chromosome yeast".

According to the suggestion, we added the data on the survival mode of telomerase-null multiple-chromosome yeasts in the revised manuscript. The detail information please referred to Figure 6—figure supplement 4.

2) The GT-tract stability data is considered extremely interesting, albeit a bit incomplete. Could growth characteristics, viability platings or other data you have address the issue of how frequent complete losses of these tracts occur and what this may cause? Is it possible that such major replication problems go undetected in your assay (because of inviability of the cells) and therefore skew the interpretation derived from it? In case there is no experimental data addressing this possibility directly, you would have to tone down the Discussion "the essentiality of CST is solely attributed to telomere protection" and include the caveat that one-step complete losses may not be recovered in your system

We thank the reviewers for this comments. Accordingly, we have added a few sentences to discuss this issue in the revision: “However, it should be noted that the replication of ITSs may encounter difficulties in the absence of CST (Gasparyan et al., 2009, Price et al., 2010, Stewart et al., 2012, Wang et al., 2014, Wang et al., 2019), which leads to cell death and might not be recovered in this system. Additionally, or alternatively, the frequency of the expansion/contraction of ITSs in *cst* mutants was too low to be detected within ~100 rounds of replication (Aksenova et al., 2015).” In addition, we have toned down the Discussion: “the essentiality of CST is mainly due to telomere protection” in the revised manuscript.

3) Role of Pol32 in telomere fusion (Figure 4E): Pol32 appears to affect the chromosome fusion rate in cdc13Δ cells. This discrepancy was likely attributed to different telomere structures between SY14 tlc1Δ and cdc13Δ cells. The reviewers thought that this speculative idea would need some more direct evidence. Otherwise, it would be better to say "we speculate different telomere structures between SY14 tlc1Δ and cdc13Δ cells".

According to this comment, we have revised the statement: “We speculate that this discrepancy could likely be attributed to different telomere structures between SY14 *tlc1*Δ and *cdc13*Δ cells” in the revised manuscript.